# *Heligmosomoides bakeri* and *Toxoplasma gondii* co-infection leads to increased mortality associated with changes in immune resistance in the lymphoid compartment and disease pathology

Edina K. Szabo[1,2¤], Christina Bowhay[1,2], Emma Forrester[1,2], Holly Liu[1,2], Beverly Dong[1,2], Aralia Leon Coria [1,2], Shashini Perera[1,2], Beatrice Fung[1,2], Namratha Badawadagi[1,2], Camila Gaio[1,2], Kayla Bailey [1,2], Manfred Ritz[1,2], Joel Bowron[1,2], Anupama Ariyaratne[1,2], Constance A. M. Finney [1,2]*

1 Department of Biological Sciences, Faculty of Science, University of Calgary, Calgary, Alberta, Canada,
2 Host-Parasite Interactions Research Training Network, University of Calgary, Calgary, Alberta, Canada

¤ Current address: Department of Cancer Immunology, Institute for Cancer Research, Oslo University Hospital, Oslo, Norway
* constance.finney@ucalgary.ca

**Data Availability Statement:** All relevant data are within the paper and its Supporting Information files.

## Abstract

Co-infections are a common reality but understanding how the immune system responds in this context is complex and can be unpredictable. *Heligmosomoides bakeri* (parasitic roundworm, previously *Heligmosomoides polygyrus*) and *Toxoplasma gondii* (protozoan parasite) are well studied organisms that stimulate a characteristic Th2 and Th1 response, respectively. Several studies have demonstrated reduced inflammatory cytokine responses in animals co-infected with such organisms. However, while general cytokine signatures have been examined, the impact of the different cytokine producing lymphocytes on parasite control/clearance is not fully understood. We investigated five different lymphocyte populations (NK, NKT, γδ T, CD4+ T and CD8+ T cells), five organs (small intestine, Peyer's patches, mesenteric lymph nodes, spleen and liver), and 4 cytokines (IFN©, IL-4, IL-10 and IL-13) at two different time points (days 5 and 10 post *T. gondii* infection). We found that co-infected animals had significantly higher mortality than either single infection. This was accompanied by transient and local changes in parasite loads and cytokine profiles. Despite the early changes in lymphocyte and cytokine profiles, severe intestinal pathology in co-infected mice likely contributed to early mortality due to significant damage by both parasites in the small intestine. Our work demonstrates the importance of taking a broad view during infection research, studying multiple cell types, organs/tissues and time points to link and/or uncouple immunological from pathological findings. Our results provide insights into how co-infection with parasites stimulating different arms of the immune system can lead to drastic changes in infection dynamics.

**Funding:** This works was funded through CAMF's grants from the Canadian Foundation for Innovation John R Evans Leaders Fund (https://www.innovation.ca) and the Natural Sciences and Engineering Research Council of Canada (NSERC, https://www.nserc-crsng.gc.ca), as well as scholarships for AA (NSERC Create in Host Parasite Interactions, https://www.ucalgary.ca/host-parasite-interactions), NB (University of Calgary Markin scholarship, https://ucalgary.ca/registrar/finances/awards-scholarshipsand-bursaries/search-awards), KB and EF (University of Calgary PURE Scholarship, https://ucalgary.ca/registrar/finances/awards-scholarships-and-bursaries/search-awards), JB and BD (NSERC, https://www.nserc-crsng.gc.ca), CG and MR (Mitacs Globalinks Scholarships, www.mitacs.ca), SP (Alberta Graduate Excellence Scholarship, https://studentaid.alberta.ca) and EKS (UCalgary Eyes High Postdoctoral Scholarship, https://ucalgary.ca).

**Competing interests:** The authors have declared that no competing interests exist.

## Introduction

Co-infections are a common reality [1]. However, understanding the complexities which underlie their impact on host immune responses remains difficult.

Co-infection by pathogens stimulating different arms of the immune response can result in different outcomes [2], although certain traits are thought to predict the interplay between multiple infectious agents. For example, parasite infection dynamics can be predicted by the type of immune effector mechanisms required to clear infection, as well as the ability of parasites to induce immunosuppression [3]. Both these traits are defined by specific cytokine signatures. Changes in the characteristic signatures can have significant impacts on disease management. This is particularly important in the context of intestinal parasitic worm infections since these are extremely common in humans (approximately 1.8 billion infections worldwide [4]), livestock [5] and wildlife [6]. The parasites negatively impact vaccine and treatment efficacy [3, 7], which can be restored with effective anthelminthic treatment that clear the worms [8].

Here, we studied the impact of two parasites that stimulate opposing immune responses in the small intestine and the gut-associated lymphoid tissue (GALT). *Heligmosomoides bakeri* (Hb), formerly *Heligmosomoides polygyrus* [9], is a parasitic nematode (roundworm) which matures within the intestinal tissue and causes a chronic infection whereby adult worms reproduce and lay eggs in the intestinal lumen [10]. The immune response to this parasite has been well characterised [10]. To clear worms, the host stimulates a potent Th2 immune response leading to the formation of a Th2 granulomas, rich in eosinophils and macrophages, that allow the trapping/killing of larvae [11]. Antibodies and intestinal physiological responses also promote worm killing/expulsion. In conjunction, a Treg response is mounted, thought to limit any potentially immunopathological Th1 response [12]. However, in susceptible mice (C57Bl/6 mice, used here), the granulomas and relatively weak Th2 response are not strong enough to clear the worms [13]; the infection remains chronic.

*Toxoplasma gondii* (Tg) is an intracellular parasite that first infects intestinal cells, and multiple other cell types as it spreads systemically [14]. The host response to Tg is a strong inflammatory response involving IFNγ production by lymphocytes. NK cells are essential for the early control of Tg [15–17], as is the induction of CD8$^+$ T cells [18]. Similarly, Tg infection studies demonstrate that γδ T cell depleted mice are less resistant to Tg than their wildtype counterparts, due to the early IFNγ production of γδ T cells during infection [19, 20]. The role of NKT cells during Tg infection is not clear. While they play a large part in the induction and initiation of inflammatory responses, this can lead to overproduction of IFNγ, which can be detrimental for the host [21]. Others have found that NKT cells may even be directly involved in the suppression of protective immunity against Tg [22]. However, most studies published on helminth-Tg co-infections focus on IFNγ production by conventional T cells (CD4$^+$ and CD8$^+$ T cell) [23–26], with much less known about the role of other lymphocyte populations (e.g. γδ T and NKT cells). Finally, as well as a strong IFNγ response, Tg stimulates high levels of IL-10 to limit immunopathology [27].

In real-world infections, the intestinal mucosa is continually subjected to a multitude of pathogens and commensals. The interplay between the different organisms impacts the ability of the immune system to maintain a strong barrier and contain these populations. Several studies have demonstrated reduced inflammatory cytokine responses in animals co-infected with an intestinal nematode and a microparasite such as Tg [24–26, 28–30]. While general cytokine signatures have been examined, the impact on the different cytokine producing lymphocytes on parasite control/clearance is not fully understood. The GALT, including the mesenteric lymph nodes (MLN), Peyer's Patches (PP) and the small intestine (SI), represents

the largest lymphatic mass in the body. The ability of lymphocytes to traffic to and interact at these sites is critical to host defence [30]. To shed more light on the lymphocyte response in the context of HbTg co-infection, we investigated the changes in Th1 and Th2 cytokines at the systemic level, as well as IFN©-producing lymphocytes (NK, NKT, CD4$^+$ T, CD8$^+$ T and ©δ T cells) at day 5 and day 10 post-Tg infection in different organs. Few studies have focused on the SI, which is Hb's niche and is Tg's first contact with the host, or on the PP which are intricately linked to the SI. This is important since lymphocytes generally display distinct phenotypes and functions in different organs [31].

Here, we found that co-infected animals had significantly higher mortality than either single infection. This was accompanied by changes in cytokine profiles and parasite loads which were dependent on the cell type (CD4$^+$ T, CD8$^+$ T, NK, NKT and γδ T cells), the organ (SI, PP, MLN, SPL and liver), the time-point (day 5 and 10 post-infection) and the cytokine (IFNγ, IL-4, IL-10 and IL-13) studied. These changes do not fully account for the mortality differences between the groups. However, our pathology results indicate that HbTg infected animals suffered from significant damage by both parasites in the SI (worm induced Th2 granulomas and Tg induced inflammation), which was likely a contributor to increased mortality. Our work provides insights into how co-infection with parasites stimulating different arms of the immune system can lead to drastic changes in infection dynamics. This helps explain why results from controlled single pathogen studies do not always translate well to real world scenarios.

## Methods

### Animals, parasites and infection protocols

C57BL/6 female mice, aged between 8–10 weeks old, were bred in house at the University of Calgary. Breeding pairs were originally purchased from Charles River Laboratories (Senneville, Quebec). All mice were housed in groups in plastic cages with enrichment where they could move around freely. They were under specific pathogen-free conditions. They had access to food and water at all times. Food was in the form of pellets, but also as DietGel (ClearH2O) for supplementation. All staff were trained in basic animal handling, as well as specific training on rodent gavage, anaesthesia and euthanasia. The University of Calgary's Animal Care Committee approved all experimental animal procedures and staff.

The Me49 strain of *Toxoplasma gondii* (Tg) (maintained in house, original stock was a gift from Dr. Georgia Perona Wright, University of British Columbia, Canada) was maintained in male C57BL/6 mice by bimonthly passage into new animals. Cysts were obtained from the brains of these animals a month after infection. Brains of infected mice were homogenised in PBS to count tissue cysts. 20 cysts were orally administered per mouse (female C57BL/6). Animals were euthanized either 5 or 10 days post infection.

Female C57BL/6 mice were orally infected with 200 *Heligmosomoides bakeri* (Hb) infective larvae (maintained in house, original stock was a gift from Dr. Allen Shostak, University of Alberta, Canada). Animals were euthanized either 12 or 17 days post infection. Larvae were obtained from fecal cultures after approximately 8–9 days of incubation at room temperature.

For co-infected animals, mice were first orally infected with Hb for 7 days. At day 7 post-infection, animals were orally infected with 20 cysts of the Me49 strain of Tg (all other experimental animals were gavaged with PBS as a control). Mice were euthanized 5 or 10 days post Tg infection (equivalent to 12 or 17 days post Hb infection).

Each experiment used 16 animals (4 per group). Mice were monitored daily. Humane endpoints were defined as: general decline of health, respiratory distress, severe neurological signs, loss of body weight > 20% or recommendation from the facility veterinarian. Animals were

euthanised within 24 hours of humane endpoints being observed. For HbTg/Tg animals, a small number of animals were found dead before the 10 days post-Tg infection. This happened during the night of days 9–10, after a check of humane endpoints on the afternoon of day 9.

For worm counts, small intestines of Hb and HbTg infected mice were harvested and opened longitudinally. The number of adult worms present in the intestinal lumen along the length of the small intestine was counted under a dissection microscope.

## Histopathology

Small intestines, spleens and livers were isolated and fixed in 4% paraformaldehyde. Paraffin embedded sections were obtained for each tissue and stained with hematoxylin and eosin (H and E) (for eosinophil/macrophage identification in the granulomas) and/or hematoxylin and Alcian blue (for goblet cell counts in the small intestine). Sections were cut and stained with H and E by the University of Calgary's Veterinary Diagnostic Services Unit. Stained tissue sections were analysed visually using the Leica DMRB light microscope and/or the OLYMPUS SZX10 microscope according to a scoring system developed by the Finney group (S1 and S2 Figs, S1 and S2 Tables). The intestinal inflammation score was adapted from [32]. Images were captured using the QCapture and the CellSens software and processed using the ImageJ software. For eosinophil and macrophage identification, one photograph of each granuloma was taken at x400 magnification using an Olympus BX 34 microscope and the Olympus CellSens software. Cells were identified by their morphology and staining patterns.

For the small intestinal, spleen and liver scoring systems, slides were blindly evaluated by two independent researchers at x40 magnification. Three scoring parameters were developed for the spleen: follicle shape, marginal zone thickness and size of the germinal center (light zone). For the liver, we used the presence/absence of necrosis, infiltrate size, proportion of infiltrates, perivascular infiltrate characteristics. The parameters within each scheme were given an equal weighting and were combined to give a total, gross pathological score for each slide. A higher score is representative of a greater amount of observed tissue damage. The pathology scale for the small intestine was adapted from [32].

For Paneth cell counts, the total number of cells per 20 VCU (villus/crypt unit) was calculated in the proximal and distal SI for each animal. For Paneth cell degranulation, each cell was given a score of either 1 (no degranulation), 2 (partial degranulation) or 3 (full degranulation) as previously described in [33, 34]. For goblet cell counts, the average number of goblet cells from five consecutive villi was calculated for each animal in the proximal and distal SI. The villi perimeters were calculated using ImageJ.

## Serum ELISAs

Blood samples were collected using a terminal cardiac bleed. Blood was left to clot for 30 minutes and then centrifuged twice at 11, 000 g at 4˚C for 10 minutes.

IFNγ cytokine production was measured in the serum of Tg and HbTg infected, and naive mice with the Mouse DuoSet ELISA development system (R&D Systems), according to the manufacturers' instructions.

## RNA/DNA extraction and quantitative PCR

RNA and gDNA were extracted from snap frozen SPL, LIV, MLN, PP, and 4 equal sections of the SI (sections 1–4, where 1 is the most proximal section and 4 the most distal from the stomach) by crushing the tissue using a pestle and mortar on dry ice and using Trizol (Ambion, Life Technologies) following manufacturer guidelines.

To measure cytokine levels in the RNA samples, cDNA was prepared using Perfecta DNase I (Quanta), and qScript cDNA Supermix (Quanta) or iScript Reverse Transcription Supermix for RT-qPCR (Bio-Rad).

Primers for quantitative PCR were obtained from Integrated DNA Technologies (San Diego). For IFNγ TCAAGTGGCATAGATGTGGAAGAA forward, and TGGCTCTGCAGGAT TTTCATG reverse, IL-10 GTCATCGATTTCTCCCCTGTG forward, and ATGGGCCTTGTA GACACCTTG reverse, IL-4 CGAAGAACACCACAGAGAGTGAGCT forward, and GACTCAT TCATGGTGCAGCTTATC reverse, IL-13 GATCTGTGTCTCTCCCTCTGA forward, and GTCCACACTCCATACCATGC reverse, ZO-1 CGCCAAATGCGGTTGATC forward and TTTACACCTTGCTTAGAGTCAGGGTT reverse, Occludin CCAGGCAGCGTGTTCCT forward and TTCTAAATAACAGTCACCTGAGGGC reverse, IL-22 ACTTCCAGCAGCCATACATC forward and CACTGATCCTTAGCACTGACTC reverse and β-actin GACTCATCGACTCCTGCTTG forward, and GATTACTGCTCTGGCTCCTAG reverse primers were used. To confirm primer product size, all products were run on a gel. Samples were run on a Bio-Rad CFX96 real time system C1000 touch thermocycler. Relative quantification of the cytokine genes of interest was measured with the delta-delta cycle threshold quantification method [35], with β-actin for normalization. Data are expressed as a fold change relative to uninfected samples.

To quantify the *T. gondii* parasite burdens in the gDNA samples, we used a standard curve of *T. gondii* tachyzoites. *T. gondii* B1 was the target gene for the quantification of parasite in all tissues or organs [36]. B1 primers (forward: TCCCCTCTGCTGGCGAAA, reverse: AGCGTTCGTGGTCAACTATCGATT) were prepared by IDT (San Diego).

## Flow cytometry

Cell suspensions were obtained from the MLN, and LIV. For the MLN, cells suspensions were obtained through mechanical disruption of whole tissues using a pestle. Perfused liver lobes were homogenised as per MLN. Perfusion was carried out through the vena cava after mice were anaesthetised, using a peristaltic pump (Gilson Minipuls3) to circulate prewarmed PBS until a change of colour was observed in the liver. Lymphocytes were isolated from the cell suspension using a 37% and 70% Percoll gradient (GE Healthcare Bio-Sciences AB, Sweden) diluted with HBSS (with or without phenol red, Lonza, Switzerland), and resuspended in RPMI.

For *ex vivo* restimulation, cells were incubated with 50ng/mL phorbol myristate acetate and 1 µg/mL ionomycin for 6 h at 37˚C with 5% $CO_2$ in the presence of BD GolgiStop. Single cell suspensions for all tissue types were stained and analysed according to [37]. Cells were stained for: viability (APC-H7 or AF700, BD Biosciences), surface markers: CD45 (BV510), CD3 (BV605), CD4 (PerCP-Cy5.5), CD8 (BUV395), NK1.1 (BV421), γδ TCR (BV711) and intracellular markers: IFNγ (APC), and granzyme B (FITC). All antibodies were purchased from BD Biosciences apart from CD4 (PerCP-Cy5.5, BioLegend), Granzyme B (FITC, eBioscience). Cells were blocked with rat α-mouse CD16/32 (Biolegend). Cells were run on an LSRFortessa X-20 flow cytometer and data analysed using FlowJo software. For analysis, doublets and dead cells were removed from the analysis, and NK, NKT, ©δ T, CD4 T and CD8 T IFN© and GZMB producing cells were quantified (S3 Fig).

## Statistical analysis

Graphpad Prism software (La Jolla, CA, USA) was used for all statistical analysis. To compare multiple groups (three or more), we performed a normality test (D'Agostino and Pearson test, unless N too small, in which case Anderson-Darling or Shapiro-Wilk). ANOVA or Kruskal-Wallis tests were performed on parametric/non-parametric pooled data, and when significant,

Sidak's/Dunn's Multiple comparisons were performed on Hb vs. HbTg and Tg vs. HbTg. For survival analysis, we used a Mantel-Cox test. Data are presented as median and individual data points, unless otherwise specified.

## Results

### Increased mortality of HbTg infected mice correlates with increased intestinal worm burden and MLN Tg burden

Hb infective larvae burrow into the submucosa of the small intestine, moult and develop into adults before they start to emerge into the intestinal lumen from day 7 post infection. At this time point, Hb specific Th2 responses (IL-4 and IL-13) are already measurable in the MLN, and Treg numbers are significantly increased [38]. To investigate how this initial infection phase impacts incoming infections, 200 *H. bakeri* larvae (L3) were orally administered to female C57Bl/6 mice 7 days prior to infection with 20 Me49 *T. gondii* tissue cysts (Fig 1A). By day 16, over 60% (8/12) Tg-infected mice survived infection in contrast to fewer than 20% (2/12) HbTg-infected animals (Fig 1B).

To determine whether decreased survival was associated with increased parasite loads, we measured Hb and Tg levels. Tg and Hb both infect their host via the small intestine. Hb has an entirely enteric lifecycle. Larval stages develop within the intestinal tissue resulting in the

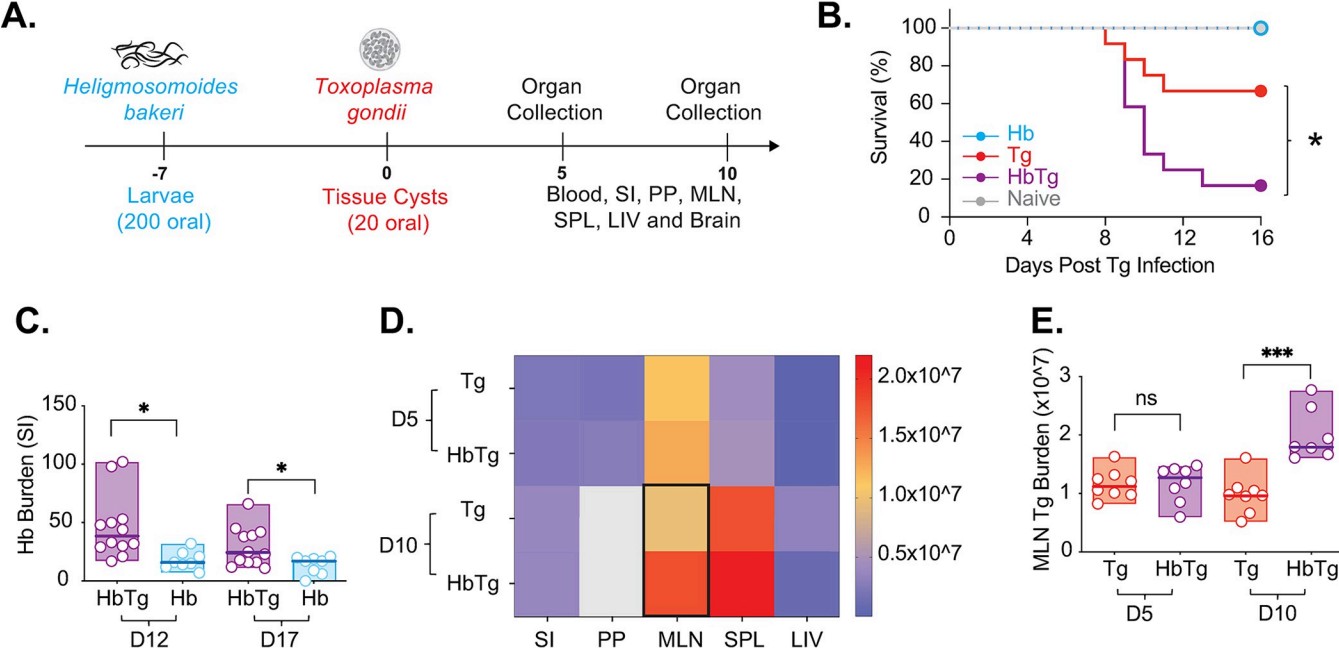

**Fig 1. *H. bakeri* co-infection leads to high mortality and increased parasite loads in mice co-infected with *T. gondii*.** (A) 200 Hb larvae were given orally to mice 7 days prior to infection with 20 Me49 Tg tissue cysts. Animals were euthanised 5 and 10 days post Tg infection (12 and 17 days post Hb infection). (B) Survival was measured for 16 days post Tg infection. N = 4 mice (Tg and HbTg-infected groups) and 2–4 mice (naïve/Hb-infected groups) per experiment, 3 independent experiments. Mantel-Cox test performed for pooled data, *p = 0.02 (Tg vs. HbTg). (C) Adult worms were counted in the small intestine of Hb and HbTg infected mice at 12 and 17 days post Hb infection. (D and E) Tg loads were measured by quantitative PCR in the small intestine (SI), Peyer's patches (PP), mesenteric lymph nodes (MLN), spleen (SPL) and liver (LIV) at 5 and 10 days post-Tg-infection. Grey boxes depict values that are out of range (>2.0*10^7). Black outline depicts significant differences between Tg and HbTg groups. For (C-I), N = 2–6 mice per group per experiment, 2 independent experiments. All data were tested for normality. Bar represents median and box delineates minimum and maximum values. (C and E) Unpaired T-tests/Mann Whitney tests were performed on parametric/non-parametric pooled data on either Hb vs. HbTg or Tg vs. HbTg, (D) ANOVA or Kruskal-Wallis tests were performed on parametric/non-parametric pooled data for N/Tg/HbTg/Hb, and when significant, Sidak's/Dunn's Multiple comparisons were performed on Tg vs. HbTg and HbTg vs. Hb; n.s. = non significant, * = p<0.05, *** = p<0.001 and **** = p<0.0001.

formation of granulomas (accumulation of immune cells around the worm), which are visible for weeks even after the worms have been cleared. Adult stages (found at days 12 and 17 post-infection) reside in the intestinal lumen [39]. At both time points, we found an increase in adult worms in the lumen of HbTg compared to Hb infected mice (Fig 1C).

Unlike Hb, Tg infects its host through the small intestine, and quickly disseminates throughout the body. This can occur as early as 3 days post-infection [14, 40, 41]. We measured Tg levels in the SI, PP, MLN, SPL and LIV at both 5 and 10 days post Tg infection. Tg was detected in all organs, at both time points in Tg and HbTg infected animals. At day 5 post Tg infection, the organ with the highest level of Tg were the MLN (Fig 1D). No difference in Tg burden between Tg and HbTg infected animals was observed at this time point in any of the organs (Fig 1D and 1E). However, 10 days post Tg infection, levels in all three lymphoid organs (PP, MLN and SPL) were highest (Fig 1D) and Tg levels were increased in the MLN of HbTg compared to Tg infected mice (Fig 1E).

Commonly Me49 infection culminates in brain tissue being invaded by approximately 3 weeks post-infection. No tissue cysts were found in any of the infected mice at d5 or day 10 post-infection, or at the time of death (up to day 16 post Tg infection).

## Cytokine profiles differ between Tg and HbTg infected mice in the MLN

We measured cytokine levels in the MLN to assess whether they were associated with the increased Tg burden measured on day 10 post infection in HbTg compared to Tg infected animals.

Th1 responses, specifically IFNγ produced by NK and CD8[+] T cells, are crucial for protection against Tg [15, 42–47]. A lack of early IFN© leads to increased Tg replication [48, 49]. At 5 days post Tg infection, HbTg animals had only approximately a third of the *Ifn©* expression measured in the Tg-infected group (Fig 2A). By 10 days post Tg infection, levels in Tg animals had decreased and the difference between the two groups was no longer apparent (Fig 2A). We also measured *Ifnγ* expression levels in the organs where we measured Tg burden (Fig 2B). Since Tg is mostly associated with pathology in the ileum [50, 51], but has also been found to replicate along the entire SI (both proximal and distal [14]), and Hb adult worms are mostly found in the proximal part of the SI (in close proximity to the stomach [52]), we analysed gene expression in the SI, divided into four equal parts (SI1-4, from proximal to distal). As expected, levels of *Ifnγ* expression in naïve and Hb infected animals were low, but elevated in the Tg infected animals (Fig 2B). The MLN were the organ with the highest fold increase (approximately 700, 5 days post Tg infection, Fig 2B) and the only site with a significant change in expression between Tg and HbTg infected animals.

The production of Th2 cytokines can lead to reduced inflammatory responses [53, 54], with IL-4 production directly inhibiting IFN© production [55]. The Th2 cytokines IL-4, IL-13 and IL-10 are involved in worm expulsion [13, 56–58]. In the absence of IL-4 and IL-13, worm expulsion is compromised [58] and high levels of IL-10 have been associated with resistant phenotypes [13]. IL-10 expression is also required in the SI to prevent necrosis and mortality of Tg infected animals [59]. We measured the gene expression profiles of *Il-4*, *Il-13* and *Il-10* (Fig 2C–2H). Unsurprisingly, we found elevated *ll-4* and *Il-13* expression in the HbTg compared to Tg infected animals at day 5 post Tg infection in the MLN (Fig 2C and 2E). This increase was still apparent for *Il-13* 10 days post Tg infection (Fig 2E). *Il-10* expression levels did not differ between the two groups at either time point (Fig 2G).

As expected, levels of *Il-4* and *Il-13* were increased in animals infected with Hb (Hb and HbTg) but not in animals infected with Tg (Fig 2D and 2F). In general, the *Il-4* and *Il-13* gene expression profiles of co-infected animals were more similar to Hb than Tg infected animals at day 5 post Tg infection. Few differences were observed between the HbTg and Hb infected

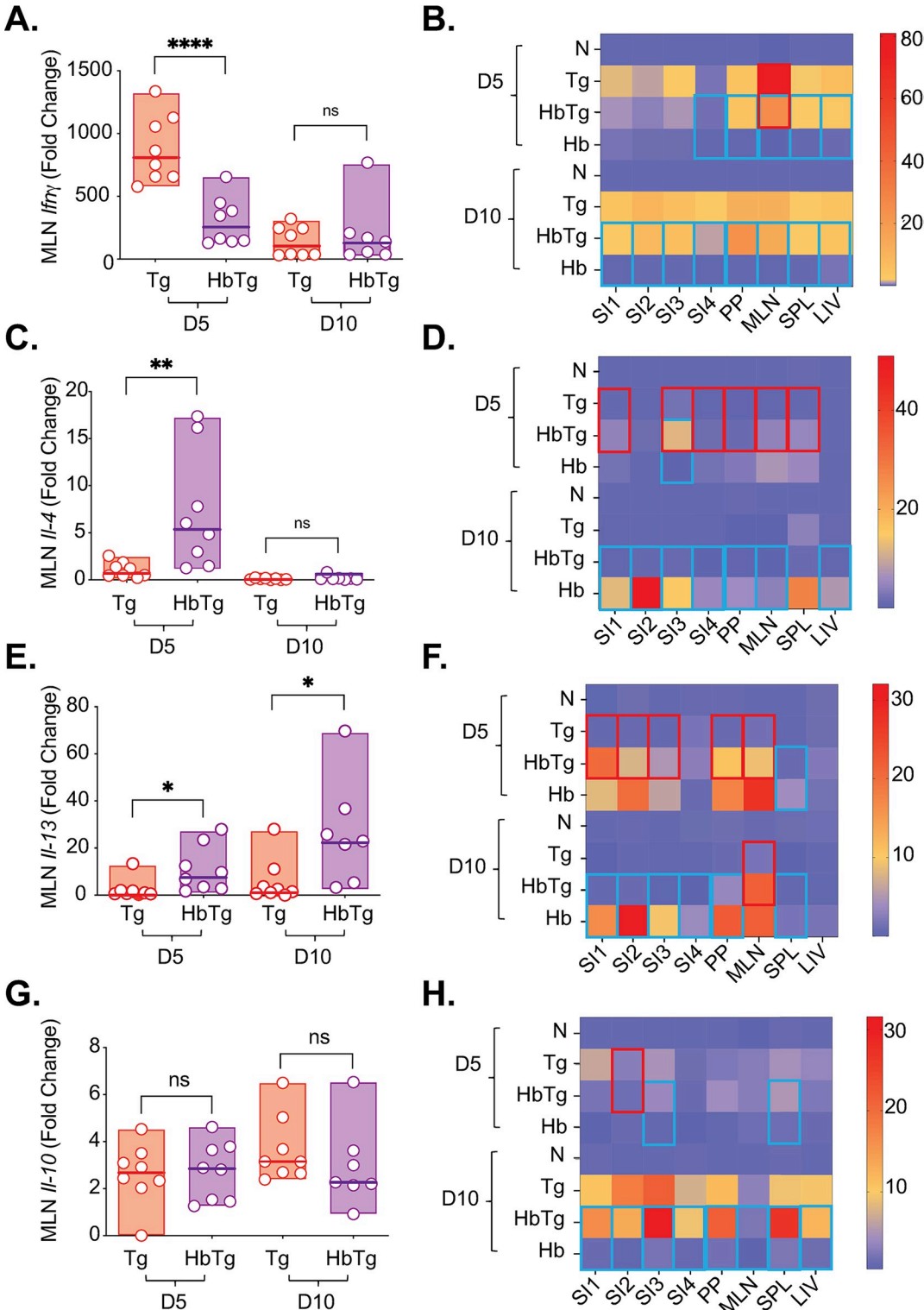

**Fig 2. The MLN gene expression profile in co-infected mice differs from that in *T. gondii*-infected mice.** 200 Hb larvae were given orally to mice 7 days prior to infection with 20 Tg tissue cysts. Gene expression was measured by quantitative RT-PCR at 5 and 10 days post-Tg-infection for *Ifnγ* (A), *Il-4* (C), *Il-13* (E) and *Il-10* (G) in the mesenteric lymph nodes (MLN). Heat maps represent median levels of *Ifnγ* (B), *Il-4* (D), *Il-13* (F) and *Il-10* (H) in the small intestine (SI, regions 1–4 from duodenum to ileum), Peyer's patches (PP), MLN, spleen (SPL) and liver (LIV) at 5 and 10 days post-Tg-infection. Red outline depicts significant differences between Tg and HbTg and blue outline depicts significant differences between HbTg and Hb. N = 2–4

mice per group per experiment, a minimum of 2 independent experiments. Data was tested for normality. ANOVA or Kruskal-Wallis tests were performed on parametric/non-parametric pooled data for N/Tg/HbTg/Hb, and when significant, Sidak's/Dunn's Multiple comparisons were performed on Tg vs. HbTg and HbTg vs. Hb; n.s. = non significant, * = $p<0.05$, ** = $p<0.01$ and **** = $p<0.0001$.

groups at this time point, despite differences in worm burden (Fig 1C). In HbTg compared to Hb infected mice, levels of *Il-4* were only increased in the SI I3 (Fig 2D) while levels of *Il-13* were only decreased in the SPL (Fig 2F). When comparing Tg and HbTg infected groups, *Il-4* and *Il-13* production were increased in at least 3 of the 4 sections of the SI, as well as in the PP and MLN (Fig 2D and 2F). For *Il-4*, levels were also increased in the SPL (Fig 2D). At day 10 post Tg infection, the situation is reversed. *Il-4* levels were reduced in HbTg compared to Hb infected animals in all organs studied but the SPL (Fig 2D). A very similar profile was observed for *Il-13* gene expression where levels were reduced in all organs studied but the MLN and LIV (Fig 2F). When comparing HbTg vs. Tg infected animals at this time point, only one difference was observed: increased *IL-13* levels in the MLN of HbTg infected animals.

## In the MLN, decreased IFNγ production is observed across five cell types but is not associated with decreased granzyme B production

The MLN are the draining lymph nodes to the SI. Different lymphocytes produce IFNγ within this immune organ, including NK and CD8+ T cells, traditionally associated with IFNγ production during Tg infection [16, 17, 60, 61], as well as NKT, γδ and CD4+ T cell more recently implicated in the inflammatory response to Tg infection [22, 62, 63]. To determine whether a particular cell type was associated with the decreased IFNγ production observed in HbTg in the MLN at 5 days post-Tg infection, we studied cell kinetics and IFNγ production in these five cell types. IFNγ-producing cells are mainly observed in the Tg-infected group (Fig 3A). γδ T and NK cells have the highest proportion of IFNγ producers with a median of approximately 12% each. Across all cell types, the percentage IFNγ-producing cells is similar between HbTg and Hb-infected animals (Fig 3A), and decreased compared to Tg-infected animals (Fig 3A–3H). By 10 days post-Tg infection, the proportion of all IFNγ producing lymphocytes studied in the MLN of HbTg infected animals were no longer decreased compared to those in Tg-infected mice (S4 Fig).

Interestingly, Granzyme B production did not mirror IFNγ production (GZMB, Fig 4). GZMB, a serine protease that activates apoptosis, has traditionally been associated with NK and CD8+ T cell killing mechanisms, explaining its increase during Tg infection [64]. However, the extent of its role during Tg infection remains controversial. Some studies have shown that Tg can actively inhibit GZMB in infected cells [65] implying that GZMB production negatively impacts Tg infection. Others have found GZMB to have a limited role in host protection [66]. GZMB is also upregulated during nematode infections, although its precise role (harmful/beneficial to the host) also remains controversial [67]. We observed no change in the proportion of GZMB-producing lymphocytes between the HbTg and Tg infected groups in any of the cell types, with NK cells having the largest proportion of GZMB producers (Fig 4A–4F). We did observe an increase in GZMB production (GZMB MFI) in the NK, CD4+ T and CD8+ T cells of Tg and HpTg infected animals compared to naïve animals; differences between Tg and HpTg groups were only apparent for the CD4+ and CD8+ T lymphocytes (Fig 4G–4L).

## In the PP, changes in IFNγ and granzyme B production are restricted to the CD8+ and CD4+ T cells

The PP are loosely organised lymphoid follicles found along the SI that exhibit differential responses after Hb infection [68]. Tg is also known to infect and replicate within them [69].

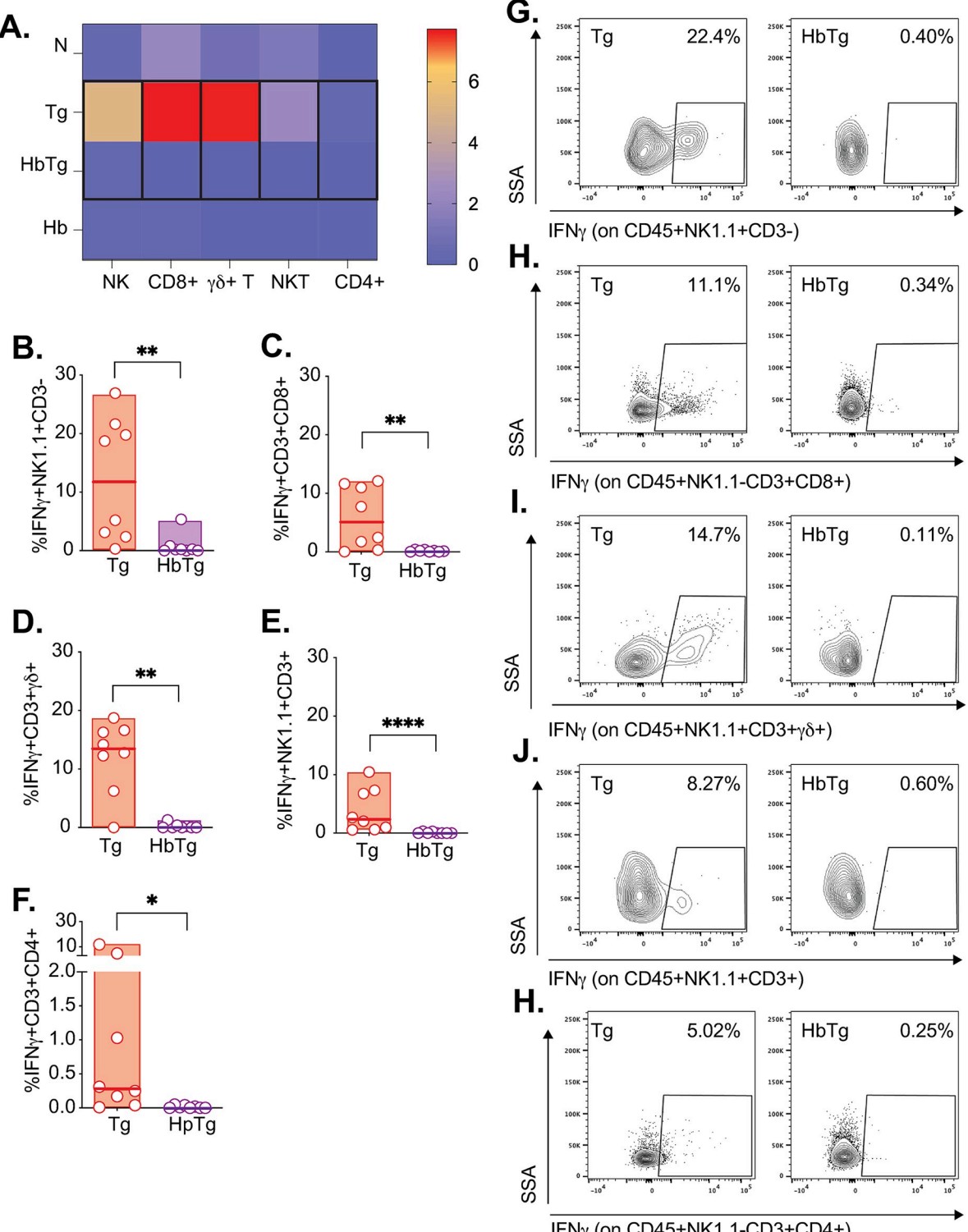

**Fig 3. Co-infected mice have reduced levels of MLN IFNγ-producing NK, CD8+ T, γδ T, NKT and CD4+ T cells.** 200 Hb larvae were given orally to mice 7 days prior to infection with 20 Tg tissue cysts. (A) Heat map represents median levels of IFN© producing cells in the MLN at 5 days post-Tg-infection. Black outline depicts significant differences between Tg and HbTg. The cell percentage of IFN©-producing (B) NK (IFN©+NK1.1+CD3-), (C) CD8 T (IFN©+CD3+CD8+), (D) ©δ T (IFN©+ CD3+©δ+), (E) NKT (IFN©+NK1.1+CD3+) and (F) CD4 T cells (IFN©+CD3+CD4+) in Tg and HbTg animals. (G-H) Flow plots depicting the IFN©+ cells in Tg and HbTg animals. N = 2–4 mice per group per experiment, 2 independent experiments. Data was tested for normality. ANOVA or Kruskal-Wallis tests were

performed on parametric/non-parametric pooled data including N/Tg/HbTg/Hb groups, and when significant, Sidak's/Dunn's Multiple comparisons were performed on Hb vs. HbTg and Tg vs. HbTg; n.s. = non significant, * = $p<0.05$, ** = $p<0.01$ and **** = $p<0.0001$.

Unlike in the MLN, we found no difference between the levels of *Ifnγ* transcripts in the Tg and HbTg groups ([Fig 2B]). However, different levels of production by the different lymphocyte populations could account for this. We therefore studied the percentage of IFN©-producing lymphocytes within the PP (NK, CD8$^+$ T, NKT, ©δ T and CD4$^+$ T). Only the IFN©-producing CD8$^+$ T cells were decreased in the HbTg group compared to the Tg group at 5 days post Tg infection ([Fig 5A–5C]). Unlike in the MLN, GZMB-producing CD8$^+$ T cells were increased in the HbTg compared to the Tg infected groups ([Fig 5D–5F]). And, we also observed an increase in GZMB production (GZMB MFI) in the NK and ©δ (but not the αβ CD4$^+$ and CD8$^+$) T cells

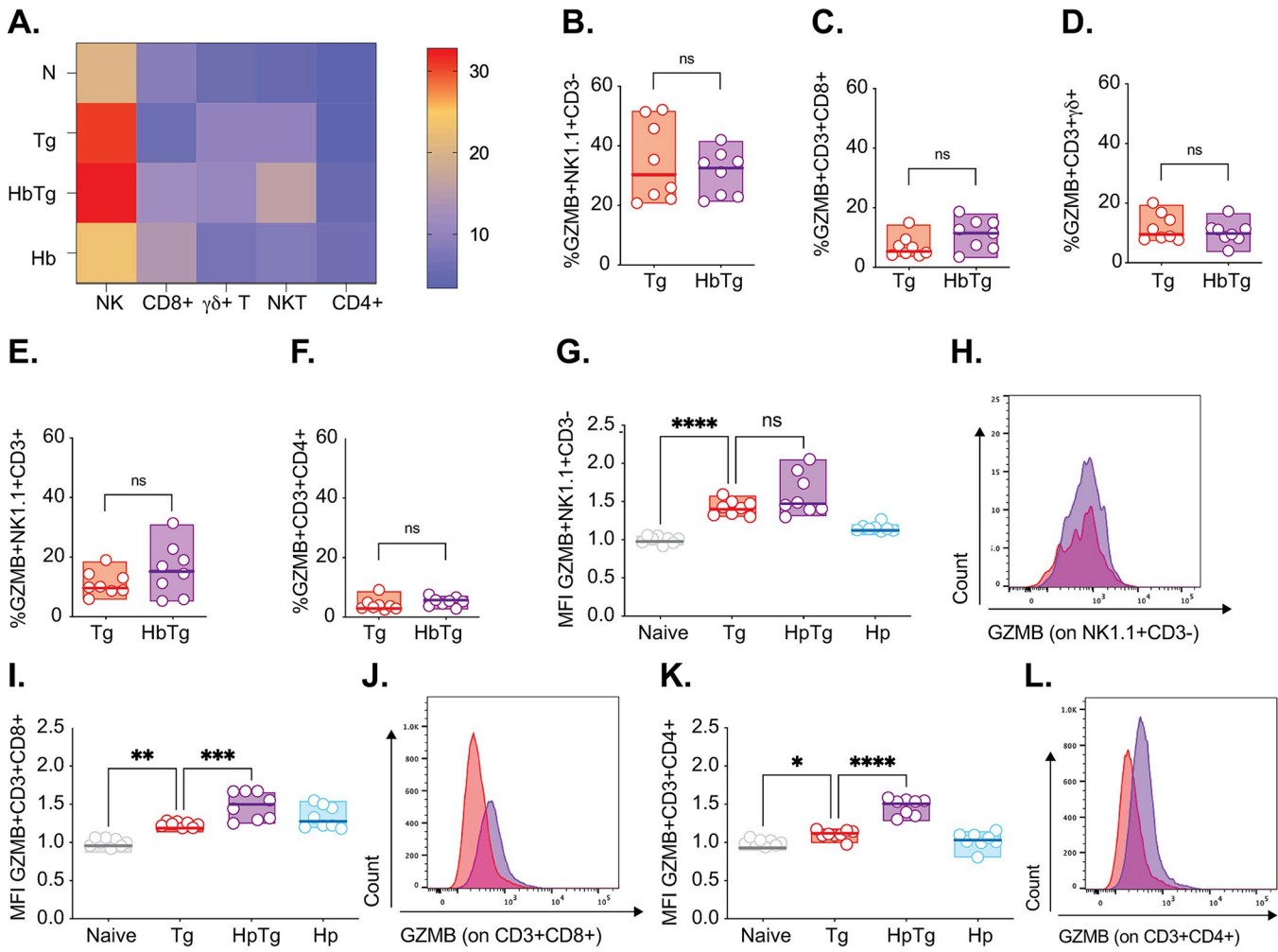

**Fig 4. Co-infected mice have increased CD8$^+$ T and CD4$^+$ T cell GZMB production in the MLN.** 200 Hb larvae were given orally to mice 7 days prior to infection with 20 Tg tissue cysts. (A) Heat map represents median levels of GZMB producing cells in the MLN at 5 days post-Tg-infection. The cell percentage of GZMB -producing (B) NK (GZMB$^+$NK1.1$^+$CD3$^-$), (C) CD8 T (GZMB$^+$CD3$^+$CD8$^+$), (D) ©δ T (GZMB$^+$CD3$^+$©δ$^+$), (E) NKT (GZMB$^+$CD3$^+$NK1.1$^+$) and (F) CD4 T cells (GZMB$^+$CD3$^+$CD4$^+$) in Tg and HbTg animals. GZMB MFI levels on (G) NK (NK1.1$^+$CD3$^-$), (I) CD8 T (CD3$^+$CD8$^+$) and (K) CD4 T (CD3$^+$CD4$^+$) cells relative to naïve animals. Flow histograms depicting GZMB MFI levels on (H) NK (NK1.1$^+$CD3$^-$), (J) CD8 T (CD3$^+$CD8$^+$) and (L) CD4 T (CD3$^+$CD4$^+$) in Tg and HbTg animals. N = 2–4 mice per group per experiment, 2 independent experiments. Data were tested for normality. ANOVA or Kruskal-Wallis tests were performed on parametric/non-parametric pooled data including N/Tg/HbTg/Hb groups, and when significant, Sidak's/Dunn's Multiple comparisons were performed on Naïve vs. Tg and Tg vs. HbTg; n.s. = non significant, * = $p<0.05$, ** = $p<0.01$, *** = $p<0.001$ and **** = $p<0.0001$.

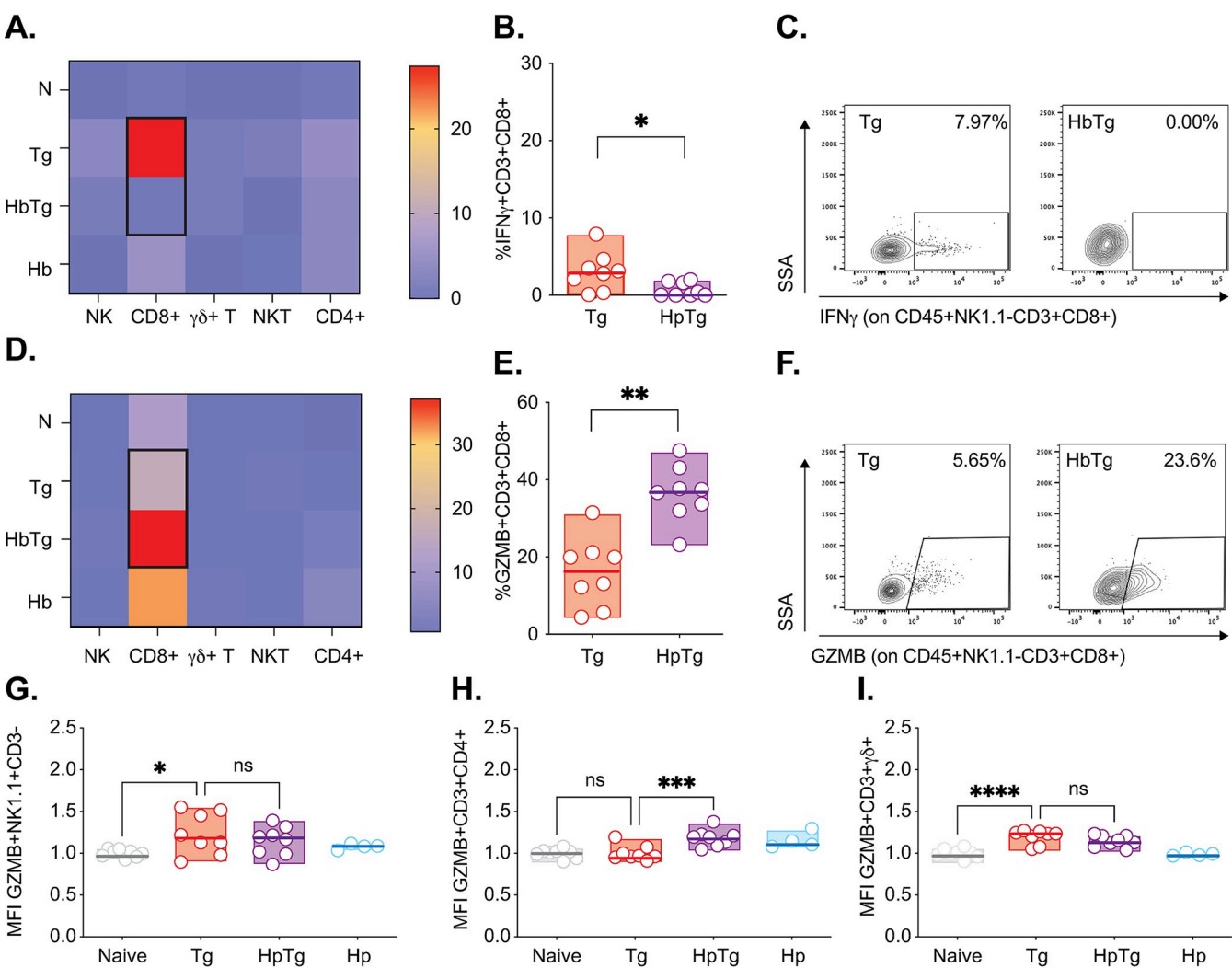

**Fig 5. Co-infected mice have reduced levels of IFNγ-producing CD8⁺ T and increased levels of MLN GZMB-producing CD8⁺ T in the PP 5 days post Tg infection.** 200 Hb larvae were given orally to mice 7 days prior to infection with 20 Tg tissue cysts. (A) Heat map represents median levels of IFN© producing cells in the MLN at 5 days post-Tg-infection. (B) The cell percentage of IFN© producing CD8 T (IFN©⁺CD3⁺CD8⁺) in Tg and HbTg animals. (C) Flow plots depicting the IFN©⁺ cells in Tg and HbTg animals. (D) Heat map represents median levels of GZMB producing cells in the MLN at 5 days post-Tg-infection. (E) The cell percentage of GZMB producing CD8 T (GZMB⁺CD3⁺CD8⁺) in Tg and HbTg animals. (F) Flow plots depicting the GZMB⁺ cells in Tg and HbTg animals. GZMB MFI levels on (G) NK (NK1.1⁺CD3⁻), (H) CD4 T (CD3⁺CD4⁺) and (I) ©δ T (CD3⁺©δ⁺) cells relative to naïve animals. N = 2–4 mice per group per experiment, 2 independent experiments. Data were tested for normality. ANOVA or Kruskal-Wallis tests were performed on parametric/non-parametric pooled data including N/Tg/HbTg/Hb groups, and when significant, Sidak's/Dunn's Multiple comparisons were performed on Naive vs. Tg and Tg vs. HbTg; n.s. = non significant, * = $p<0.05$, ** = $p<0.01$, *** = $p<0.001$ and **** = $p<0.0001$.

of Tg and HpTg infected animals compared to naïve animals; differences between Tg and HpTg groups were only apparent for the CD4⁺ T lymphocytes (Fig 5G–5I). By day 10 post Tg infection, no differences were observed in the PP between the HbTg and Tg groups in any of the parameters studied (S5 Fig). Differences between the HbTg and Tg groups were not found in any of the IFN©-producing or the GZMB-producing lymphocytes in the spleen and/or liver at either time point (S6 and S7 Figs).

Cell numbers, rather than subset percentages, can provide useful insights when studying organs that change size during infection. Compared to the naïve group, MLN cell numbers were increased approximately 4–5 times in all infected groups; no difference in cell number was observed between infection groups (S8 Fig). We did not study the cell numbers for the PP,

as they are not a discreet organ and the number of cells varies per experiment. The numbers of NK, NKT, ©δ T and CD4$^+$ T cells, but not CD8 T cells, were increased in the MLN of the Tg infected compared to co-infected group (S9A–S9E Fig). This translated into a decrease in the total number of IFN©$^+$ CD8$^+$ T, NKT, ©δ T and CD4$^+$ T cells (S9F–S9J Fig) and an increase in the number of GZMB$^+$ NK, NKT and CD4$^+$ T cells (S9K–S9O Fig).

## HbTg infected animals have intestinal pathology characteristic of both Tg and Hb parasites

Increased Tg burden in the MLN of HbTg infected mice is associated with decreased levels of IFN© by all IFN©-producing lymphocytes. However, these transient, localised changes alone are unlikely to account for the difference in mortality between the HpTg and Tg infected animals. We therefore investigated whether the increased mortality and altered cytokine levels observed in co-infected animals correlated with increased pathology. We focused first on the SI, and particularly the proximal region, since this is a preferred niche for both Hb and Tg parasites.

Whole small intestines were formalin fixed, paraffin embedded and stained with hematoxylin and eosin to investigate intestinal pathology. Gene expression of tight junction proteins *Zo1* and *Occludin*, and the key regulator of inflammation *Il-22* were measured in the SI (SI2 region). Epithelial barrier dysfunction is characterised by decreased levels of *Zo1* and *Occludin*, which has been observed during Tg infection both at the gene expression [70] and protein levels [71]. Interestingly, lower *Occludin* levels have been associated with facilitating Tg infection [71] as well as protecting epithelial cells from infection [72]. Transgenic animals lacking *Il-22* infected with Tg have higher mortalities than their wildtype counterparts [73]. As expected, inflammation score (as measured by cell infiltrates and lack of epithelium integrity) and *Il-22* levels increased with Tg infection while *Zo1* and *Occludin* levels decreased. We saw no significant difference between Tg and HbTg groups in these parameters at day 5 (Fig 6A–6E and S1 Table) post Tg infection, prior to the start of mortality (Fig 1B), however inflammation score (Fig 6A) and *Il-22* (Fig 6E) of most animals with HbTg infection were similar to Hb mice compared to Tg alone. Paneth cells have recently been shown to play an important role in limiting immunopathology and microbiota dysbiosis driven by Tg infection [74, 75]. Again, we found no difference between Tg and HbTg animals in Paneth cell numbers or their degranulation state at 5 days post Tg infection (Fig 6F–6H). The only differences we observed between Tg and HbTg infected animals in the SI at 5 days post Tg infection were the intestinal villi size, the number of goblet cells, the weight of the intestinal tissue and is the presence of granulomas. We, like others [76], observe villus atrophy (decreased length) in the proximal SI during Tg infection (Fig 6I). Goblet cells serve an important physical barrier against pathogens through the maintenance and production of the mucus layer in the small intestine through the production of mucins [77]. Their increase in number (Fig 6J and 6K), and the resulting increase in intestinal weights (Fig 6L), coupled to granuloma formation (Hb-induced characteristic influx of macrophages and eosinophils, Fig 6M–6Q), are all associated with Hb infection [78]. These changes are also observed in the HbTg, but not the Tg animals at 5 days post Tg infection (Fig 6J–6L).

Once mortality differences appear between Tg and HbTg infected animals, by 10 days post Tg infection, the only remaining differences between groups are the reduction in Paneth cell numbers and the presence of granulomas (Fig 7A–7N). Interestingly, despite the granulomas being the same size between HbTg and Hb infected groups (Fig 7L), there is a significant reduction in eosinophils in the granulomas from HbTg infected animals (Fig 7M), which could indicate an influx of Th1 rather than Th2 cells.

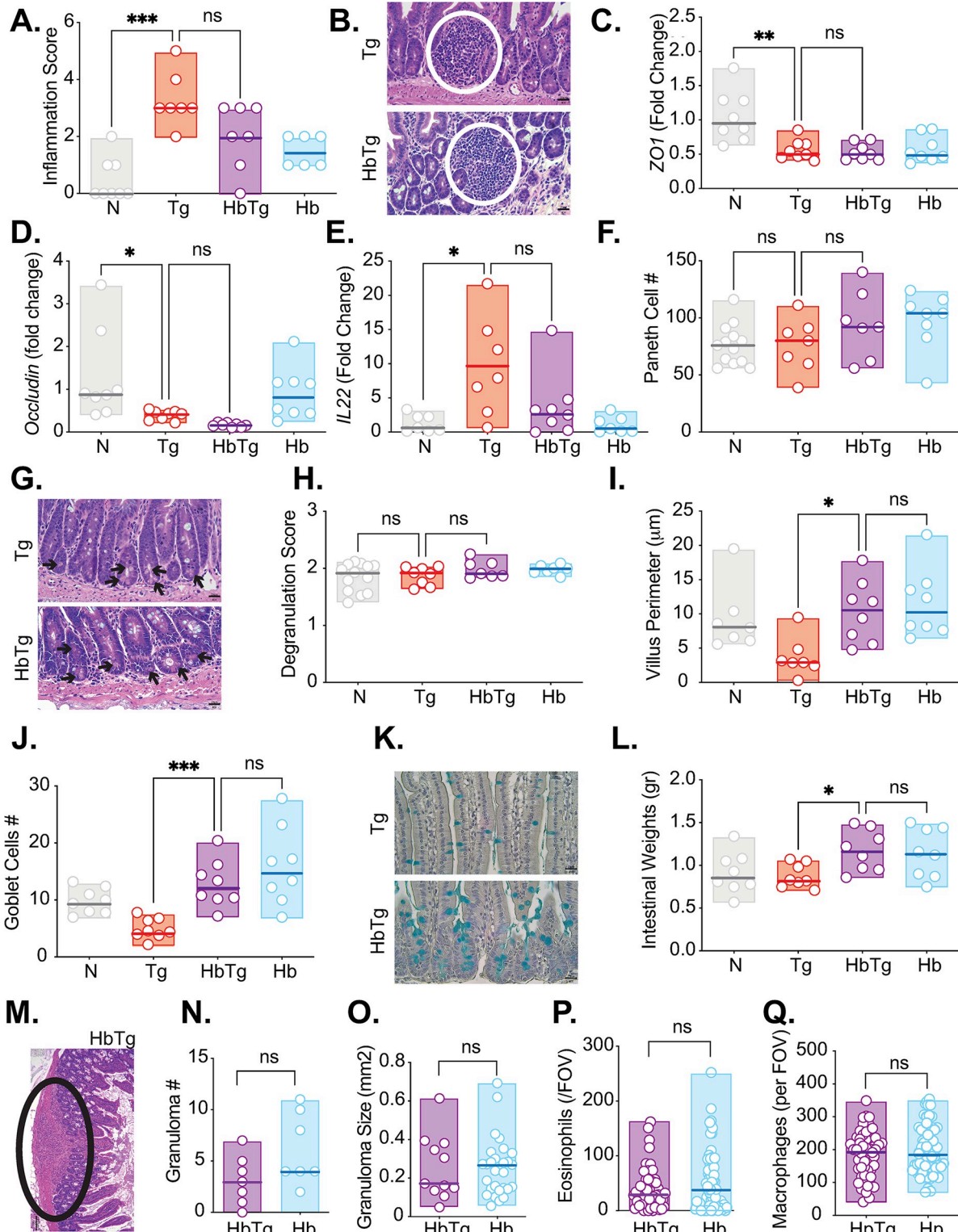

**Fig 6. Pathology in the small intestine differs between *T. gondii* and co-infected animals at 5 days post Tg infection.** 200 Hb larvae were given orally to mice 7 days prior to infection with 20 Tg tissue cysts. Small intestines were harvested from mice 5 days post Tg infection, and formalin fixed. 6 μM slides were cut from the paraffin embedded swiss rolls, and stained with hematoxylin and eosin. (A) Average small intestine inflammation pathology score for each mouse. (B) Representative images from Tg and HbTg animals. Scale: 20 μm. White circles depict leukocyte infiltrates (Tg pathology). Gene expression was measured by quantitative RT-PCR for *Zo1* (C), *Occludin* (D) and *Il-22* (E)

in SI2 of the SI. (F) Average number of Paneth cells identified on H&E stained sections in 20 villi/crypt units of each experimental group. (G) Representative images from Tg and HbTg animals. Scale: 20 μm. Black arrows depict Paneth cells. (H) Average degranulation score per villi crypt unit. (I) Average villi perimeter in the proximal small intestine of each experimental group. The perimeter of 5 intact continuous villi was measured and averaged. (J) Paraffin embedded swiss rolls were stained with Alcian blue to identify goblet cells. Average number of goblet cells in 5 intact and continuous villi within the proximal small intestine of each experimental group. (K) Representative images from HbTg animals. Scale: 20 μm. Blue cells depict goblet cells. (L) Small intestines were dissected and weighed. (M) Representative image of a granuloma in HbTg mice on an H&E stained paraffin embedded section. Black circle depicts granuloma (Hb pathology). Scale: 100μm. (N) Granulomas were counted in the small intestine of Hb and HbTg infected mice using a dissection microscope. (O) Granuloma size was calculated for the granulomas of Hb and HbTg infected mice observed on H&E stained paraffin embedded formalin fixed sections. The perimeter of each granuloma was measured and the surface area calculated (a minimum of 11 granulomas measured per group). Eosinophil (P) and macrophage (Q) counts within the centre of the granulomas. H and E slides were used to identify eosinophils and macrophages at x400 magnification. Cells were counted for one field of view (FOV) per granuloma (a minimum of 55 granulomas per group). N = 2–5 mice per group per experiment, 2 independent experiments. Normality test followed by ANOVA/Kriskal-Wallis with post-tests were performed for pooled data, n.s. = non significant, * = p<0.05, ** = p<0.01, *** = p<0.001.

Increased granuloma numbers and size are associated with resistance [13]. We found no differences in granuloma number of size between HbTg and Hb infected mice. Very few granulomas containing developing worms at days 12 and 17 post Hb infection (1 out of 9 in the HbTg infected group 17 days post-infection and 4 out of 43 in the Hb infected group 12 days post-infection) were observed, suggesting that differences in worm infection kinetics were not responsible for the differences in worm burden observed between the two groups (Fig 1C).

After having replicated in the intestinal tissue, Tg rapidly disseminates to other organs, including the spleen and liver. Analysis of spleen and liver sections indicate that pathology of both organs increased over time but that Tg and HbTg infected animals had similar severe pathologies, while Hb infected animals were far less affected (S10 Fig).

## Discussion

Our data confirm the recently published work demonstrating increased mortality in animals coinfected with Tg and Hb compared to Tg alone [26]. At first, this may appear counterintuitive. The immunoregulatory effects of helminths have been intensely studied [79–83] and HbTg co-infection was previously shown to improve survival in association with a decrease in serum IFN© levels [24]. We observed no improvement in survival or change in serum IFN© levels (S11 Fig). Early IFN© production plays a key role in controlling *T. gondii* replication [46] but IL-10 is also necessary to limit pathology stimulated by the strong inflammatory response [59]. The balance of these two cytokines is key to infection outcome [27]. Surprisingly, we did not find marked differences in *Ifnγ* and *Il-10* gene expression in all organs and time points studied. Tg parasite burden was linked to *Ifnγ* levels: in tissues with high parasite numbers, we recorded high levels of *Ifnγ* (Fig 2). However, compared to Tg infected animals, HpTg infected animals only had decreased levels of MLN *Ifnγ* and SI2 *Il-10* at day 5 post Tg infection, despite increased *Il-4* and *Il-13* in the SI, PP, MLN and SPL. The local decrease in MLN *Ifnγ* was associated with increased MLN Tg burden 5 days later, highlighting the different responses in different tissues. Our data support recent findings which demonstrate that Hb impairs early Tg-induced immunity, resulting in decreased IFNg levels [26]. Whether the observed decrease was due directly to increases in IL-4 and/or IL-13 production remains to be elucidated. Interestingly, despite IL-4 and IL-13 being increased in HbTg infected animals, levels were lower than in Hb infected animals, and likely contributed to the higher worm burdens in the coinfected group.

IFN© production by CD4+ and CD8+ T cells is crucial for protective immunity and overall survival of Tg infected animals [44, 84, 85] although it is important to remember that there is a key contribution by inflammatory neutrophils and monocytes early in infection [26]. We (Figs 3 and 5A–5C) and others [24–26] have shown that HbTg co-infection results in decreased

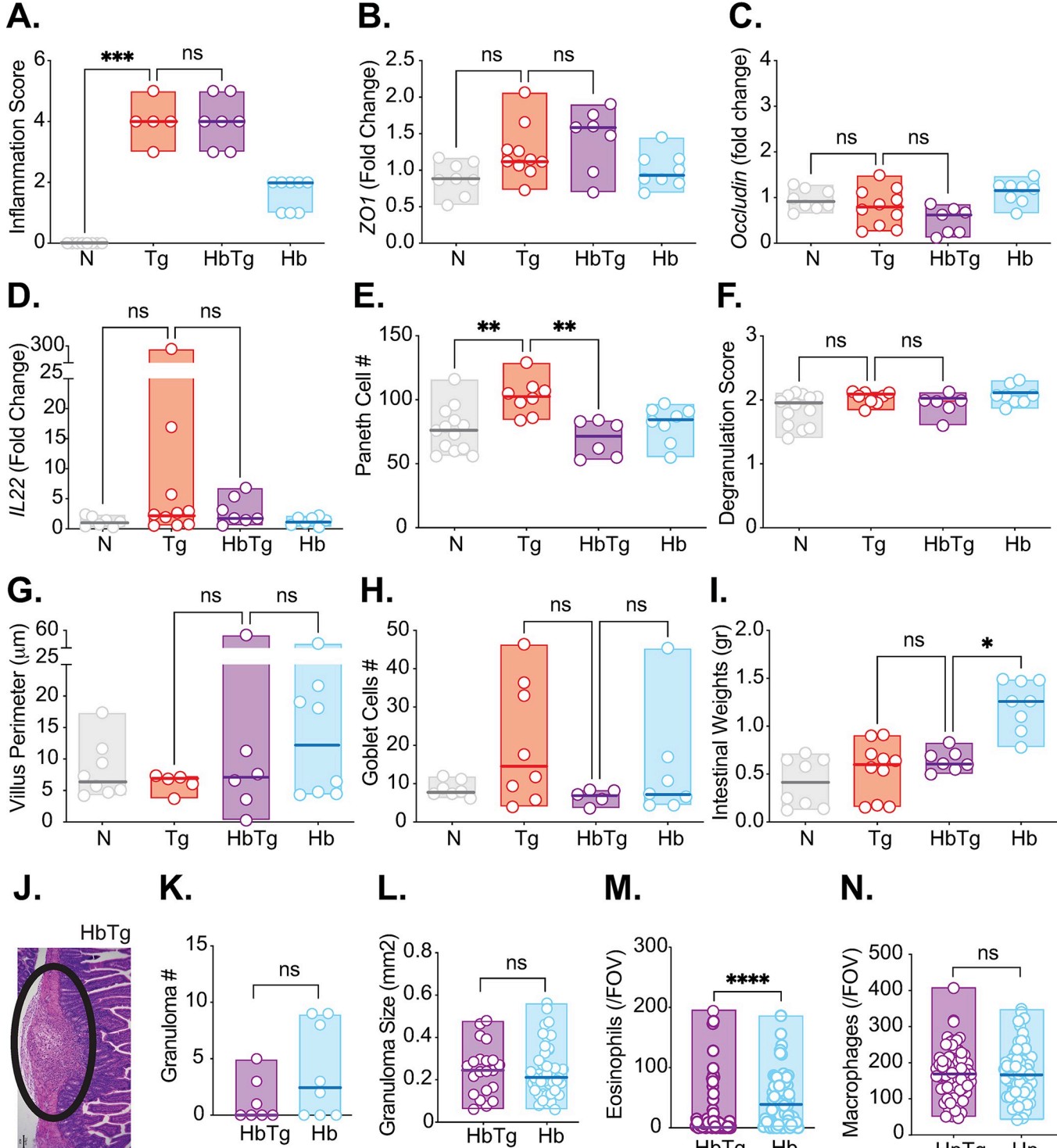

**Fig 7. Decreased Paneth cell number and the presence of granulomas differentiate HbTg from Tg animals at 10 days post Tg infection.** 200 Hb larvae were given orally to mice 7 days prior to infection with 20 Tg tissue cysts. Small intestines were harvested from mice 10 days post Tg infection, and formalin fixed. 6 μM slides were cut from the paraffin embedded swiss rolls, and stained with hematoxylin and eosin. (A) Average small intestine inflammation pathology score for each mouse. (B) Gene expression was measured by quantitative RT-PCR for *Zo1* (B), *Occludin* (C) and *Il-22* (D) in SI2 of the SI. (E) Average number of Paneth cells identified on H&E stained sections in 20 villi/crypt units of each experimental group. (F) Average degranulation score per villi crypt unit. (G) Average villi perimeter in the proximal small intestine of each experimental group. The perimeter of 5 intact continuous villi was measured and averaged. (H) Paraffin embedded swiss rolls were stained with Alcian blue to identify goblet cells. Average number of goblet cells in 5 intact and continuous

villi within the proximal small intestine of each experimental group. (I) Small intestines were dissected and weighed. (J) Representative image of a granuloma in HbTg mice on an H&E stained paraffin embedded section. Black circle depicts granuloma (Hb pathology). Scale: 100μm. (K) Granulomas were counted in the small intestine of Hb and HbTg infected mice using a dissection microscope. (L) Granuloma size was calculated for the granulomas of Hb and HbTg infected mice on H&E paraffin embedded formalin fixed sections. The perimeter of each granuloma was measured and the surface area calculated (a minimum of 11 granulomas measured per group). Eosinophil (M) and macrophage (N) counts within the centre of the granulomas. H and E slides were used to identify eosinophils and macrophages at x400 magnification. Cells were counted for one field of view (FOV) per granuloma (a minimum of 55 granulomas per group). N = 2–5 mice per group per experiment, 2 independent experiments. Normality test followed by ANOVA/Kriskal-Wallis with post-tests were performed for pooled data, n.s. = non significant, * = p<0.05, ** = p<0.01, *** = p<0.001, **** = p<0.0001.

IFN©−producing CD4+ and CD8+ T cells. NK cells are also critical for driving an effective innate response [15, 16, 42], including the development of local IL-12 producing DCs and macrophages, that drive effector responses [17, 25, 86] and both NKT [21] and ©δ T cells [22] have been implicated in early response to Tg-induced intestinal pathology. A recent study using a novel IFN© reporter mouse, demonstrated that CD4+ T, CD8+ T, ©δ T and NK cells all express IFN© during Tg infection [87]. They found that although IFN©-producing CD4+ T and CD8+ T cells were predominantly responsible for combating acute protection, IFN©-producing NK and ©δ T cell also likely played a role. Our results demonstrate that HpTg co-infection leads to a decrease in all IFN©−producing lymphocytes (CD8+ and CD4+ T as well as NK, NKT, ©δ T cells) in the MLN as a proportion (Fig 3) and in absolute numbers (S9 Fig), which helps explain the subsequent increase in Tg burden in this organ. However, we did not confirm whether lymphocytes were exhausted in our experiments. This is a limitation since it would make them unresponsive to PMA/Ionomycin stimulation and suggest that coinfection with Hb led to general lymphocyte exhaustion rather than simply a decrease in IFN© production. Future studies should consider measuring the expression of inhibitory markers such PD-1, Tim-3 and Lag-3 associated with exhaustion [88] to rule this out.

Despite the ablation of the IFN© responses in all lymphocytes in the MLN of HbTg infected animals 5 days post infection, GZMB production did not follow the same pattern (Fig 4 and S5 Fig). GZMB is a molecule involved in CD8+ T cell and NK cell cytotoxicity [19, 89] and we were surprised it was not decreased, like IFN©, in HbTg animals or associated with better control of Tg parasite loads. However, this may be because Tg itself can inhibit lymphocyte degranulation [90] and GZMB-mediated apoptosis of infected cells [65, 91], or the fact that the impact of worm infection on GZMB is unclear [67]. Studies have reported that IL-4 production results in increased cytotoxicity in splenic NK cells in the context of *Nippostrongylus brasiliensis* (parasitic nematode) infection [92]. Others have found that IL-4 suppresses NK cell cytotoxicity *in vitro* [93–95]. GZMB also inhibits tissue repair at mucosal surfaces and is being studied as a therapeutic target for wound healing [96, 97]. Elevated levels in HbTg compared to Tg animals may be contributing to tissue damage.

Aside from the temporary lack of IFN© producing lymphocytes, the observed survival differences were also linked to differences in intestinal pathology. Hb and Tg share the small intestinal niche. Tg is known for causing ileitis (inflammation of the ileum, [98]. However, pathology in the duodenum and jejunum, both niches for Hb, have also been described [76, 99]. When mice were co-infected with Tg, Hb was transitioning from a tissue dwelling to a lumen dwelling phase, resulting in pathology in the proximal small intestine. The pathology stemmed from immune effects on the intestinal epithelium (change in secretory cell populations like goblet cells), the granulomas created from an influx of immune cells to the site of infection as well as the damage caused by the worms exiting the intestinal tissue itself. Unsurprisingly we observed that HbTg infected animals had transient increases in goblet cells, *Il-4* and *Il-13* gene expression and a sustained presence of Th2 granulomas compared to Tg infected animals, all associated with Hb infection. We also observed a transient decrease in

intestinal permeability and late increases in *Ifnγ* and *Il-10* expression compared to naïve animals (Figs 2, 6 and 7). Th2 granulomas are composed mainly of M2 macrophages and eosinophils that promote fibrotic lesions [11]. In HbTg animals, at the onset of mortality, the combination of these large fibrotic granulomas with reduced Th2 cells, as well as reduced Paneth cell numbers to control intestinal inflammation stimulated by Tg, likely contributed to intestinal dysfunction and increased mortality. In support of this, we found no pathological differences between the HpTg and Tg groups within the spleen or liver (S10 Fig) or the Tg burdens in these 3 organs (Fig 1). In both Tg and HbTg infected mice, we observed severe liver and spleen pathology by 10 days post Tg infection (S10 Fig).

Previous studies have observed that animals coinfected with Hb and an inflammatory intestinal pathogen have increased mortality associated with significant weight loss approximately 10 days post-infection onwards. These include animals co-infected with Hb and Tg [26], Hb and *Citrobacter rodentium* [100] and Hb and West Nile Virus (WNV) [101]. The weight loss has been directly linked to IL-4 mediated events since treating animals with IL-4c had the same effect as an Hb infection [101] and the weight loss, and increased mortality, was reversed in STAT6 knock out animals [100, 101]. In the case of HbWNV co-infected animals, activation of a succinate-tuft cell-IL-25-IL-4R-intestinal circuit was responsible for the increased mortality and weight loss [101]. However, it was linked to intestinal pathology (reduced villus height and bacterial translocation, a read out for intestinal permeability) which we did not observe in our HbTg infected animals.

Others have found that the mechanisms underlying the weight loss observed in HpTg infected animals is unclear, and hypothesised that the increased mortality may be due to tissue stress and proinflammatory responses in certain organs [26]. The authors concluded that uncontrolled parasite replication and/or impaired T cell responses were not responsible, but that decreased food intake, poor nutrient absorption, or increased energy expenditure could be responsible for increased mortality in coinfected mice. We agree with this hypothesis. We found few differences in parasite replication and/or impaired responses despite studying two infection time points and 5 different organs. We did not measure weight loss past 10 days post infection due to the high mortality in the HbTg infected group. However, infection with Hb and *N. brasiliensis* reduces glucose absorption by the small intestine in a STAT6 dependent manner [62, 102]. And most recently, excretory/secretory products from the helminth *Schistosoma mansoni* were shown to inhibit food intake and decrease body fat mass in a STAT6-independent mechanism in animals fed a high fat diet [103].

Our results provide insights into how co-infection with parasites stimulating different arms of the immune system can lead to drastic changes in infection dynamics. Understanding the intricacies of immune responses during co-infections is key to developing models that will provide better translatability to real world situations, where invariably humans, livestock, companion animals and wildlife will be harbouring more than one type of infection.

## Supporting information

**S1 Fig. The spleen pathology scale.** (A). Follicle damage: All follicles are distinct and defined in shape (score of 1, top left). A majority of follicles are distinct and defined in shape (score of 2, top middle). Approximately equal numbers of intact and damaged follicles. (score of 3, top right). The majority of follicles are damaged, with some defined follicles observed (score of 4, bottom left). All follicles are damaged (score of 5, bottom right). Scale 10um. (B) Marginal zone thickness: More than 50% of the follicles have a thick marginal zone (score of 1, left). More than 50% of the follicles have a thin marginal zone (score of 2, middle) More than 50% of the follicles do not have an observable marginal zone (score of 3, right). (C) Size of the light

zone of the germinal centre: More than 50% of the follicles have a small germinal center (score of 1, left). More than 50% of the follicles have a large germinal center (score of 2, middle). More than 50% of the follicles have no observable germinal center (score of 3, right).
(TIF)

**S2 Fig. The liver pathology scale.** (A). Necrosis: absence (score of 1, left) and presence (score of 2, right) of necrosis. Scale 10um. (B) Infiltrate Size: All infiltrates observed are organized into small groups (score of 1, left), 75% of infiltrates observed are organized into small groups and 25% into large groups (score of 2, middle left), 50/50 split between small and large infiltrate groups (score of 3, middle right), 100% of infiltrates are organized into large groups (score of 4, right). (C). Proportion of infiltrates: very few infiltrate groups are observed (score of 1, top left), some groups of infiltrates are observed (score of 2, top middle), a medium number of infiltrates is observed (score of 3, top right), many infiltrate groups are observed and occupy the majority of the tissue (score of 4, bottom left), infiltrate groups are distinguishable, and occupy most of the tissue (score of 5, bottom middle), and Infiltrate groups are indistinguishable and occupy the entire tissue (score of 6, bottom right). (D) Perivascular infiltrates: Very few infiltrating leukocytes can be observed in the vessels (score of 1, left). Infiltrating leukocytes are observed in small numbers leaving most of the vessels (score of 2, left middle). Most of the vessels are surrounded by infiltrating leukocytes (score of 3, right middle). All of the vessels are filled with infiltrating leukocytes (score of 4, right).
(TIF)

**S3 Fig. Flow cytometry gating strategy.** Doublets were removed using the FSC-A and FSC-H parameters. Live cells were selected, followed by CD45$^+$ cells. The CD3- subset was used to identify NK cells using NK1.1. The CD3$^+$ subset was divided into ©δ cells (©δ$^+$), NKT cells (©δ-αβ$^+$NK1.1$^+$) and the αβ cells were subdivided into CD4$^+$ and CD8$^+$ cells.
(TIF)

**S4 Fig. Co-infected mice have similar levels of MLN IFNγ-producing NK, CD8$^+$ T, γδ T, NKT and CD4$^+$ T cells to Tg infected mice 10 days post Tg infection.** 200 Hb larvae were given orally to mice 7 days prior to infection with 20 Tg tissue cysts. The cell percentage of IFN©-producing (A) NK (IFN©$^+$NK1.1$^+$CD3$^-$), (B) CD8$^+$ T (IFN©$^+$CD3$^+$CD8$^+$), (C) ©δ T (IFN©$^+$CD3$^+$©δ$^+$), (D) NKT (IFN©+NK1.1$^+$CD3$^+$) and (E) CD4$^+$ T cells (IFN©$^+$CD3$^+$CD4$^+$) in Tg and HbTg animals 10 days post Tg infection. N = 2–4 mice per group per experiment, 2 independent experiments. Data was tested for normality. ANOVA or Kruskal-Wallis tests were performed on parametric/non-parametric pooled data including N/Tg/HbTg/Hb groups, and when significant, Sidak's/Dunn's Multiple comparisons were performed on Hb vs. HbTg and Tg vs. HbTg; n.s. = non significant.
(TIF)

**S5 Fig. Co-infected and Tg infected mice have similar levels of IFNγ and GZMB producing cells in the PP 10 days post Tg infection.** 200 Hb larvae were given orally to mice 7 days prior to infection with 20 Tg tissue cysts. The percentage of IFN©-producing (A) NK (IFN©$^+$NK1.1$^+$CD3$^-$), (B) CD8$^+$ T (IFN©$^+$CD3$^+$CD8$^+$), (C) ©δ T (IFN©$^+$CD3$^+$©δ$^+$), (D) NKT (IFN©$^+$NK1.1$^+$CD3$^+$) and (E) CD4$^+$ T cells (IFN©$^+$CD3$^+$CD4$^+$) in Tg and HbTg animals. The cell percentage of GZMB-producing (F) NK (GZMB$^+$NK1.1$^+$CD3$^-$), (G) CD8$^+$ T (GZMB$^+$CD3$^+$CD8$^+$), (H) ©δ T (GZMB$^+$CD3$^+$©δ$^+$), (I) NKT (GZMB$^+$NK1.1$^+$CD3$^+$) and (J) CD4$^+$ T cells (GZMB$^+$CD3$^+$CD4$^+$) in Tg and HbTg animals. GZMB MFI levels on (K) NK (NK1.1$^+$CD3$^-$), (L) CD8$^+$ T (CD3$^+$CD8$^+$), (M) ©δ T (CD3$^+$©δ$^+$), (N) NKT (NK1.1$^+$CD3$^+$) and (O) CD4$^+$ T (CD3$^+$CD4$^+$) cells relative to naïve animals. N = 2–4 mice per group per experiment, 2 independent experiments. Data were tested for normality. ANOVA or Kruskal-Wallis

tests were performed on parametric/non-parametric pooled data including N/Tg/HbTg/Hb groups, and when significant, Sidak's/Dunn's Multiple comparisons were performed on Naive vs. Tg and Tg vs. HbTg; n.s. = non significant.
(TIF)

**S6 Fig. Co-infected mice have similar levels of IFNγ and GZMB producing cells in the SPL.** 200 Hb larvae were given orally to mice 7 days prior to infection with 20 Tg tissue cysts. The cell percentage of IFNγ-producing (A) NK (IFNγ$^+$NK1.1$^+$CD3$^-$), (B) CD8$^+$ T (IFNγ$^+$CD3$^+$CD8$^+$), (C) γδ T (IFNγ$^+$CD3$^+$γδ$^+$). (D) NKT (IFNγ$^+$NK1.1$^+$CD3$^+$) and (E) CD4$^+$ T cells (IFNγ$^+$CD3$^+$CD4$^+$) in Tg and HbTg animals 5 days post Tg infection. The percentage of IFNγ-producing (F) NK (IFNγ$^+$NK1.1$^+$CD3$^-$), (G) CD8$^+$ T (IFNγ$^+$CD3$^+$CD8$^+$), (H) γδ T (IFNγ$^+$CD3$^+$γδ+), (I) NKT (IFNγ$^+$NK1.1$^+$CD3$^+$) and (J) CD4$^+$ T cells (IFNγ$^+$CD3$^+$CD4$^+$) in Tg and HbTg animals 10 days post Tg infection. The cell percentage of GZMB-producing (K) NK (GZMB$^+$NK1.1$^+$CD3$^-$), (L) CD8$^+$ T (GZMB$^+$CD3$^+$CD8$^+$), (M) γδ T (GZMB$^+$CD3$^+$γδ$^+$), (N) NKT (GZMB$^+$NK1.1$^+$CD3$^+$) and (O) CD4$^+$ T cells (GZMB$^+$CD3$^+$CD4$^+$) in Tg and HbTg animals 5 days post Tg infection. The cell percentage of GZMB-producing (P) NK (GZMB$^+$NK1.1$^+$CD3-), (Q) CD8$^+$ T (GZMB$^+$CD3$^+$CD8$^+$), (R) γδ T (GZMB$^+$CD3$^+$γδ$^+$), (S) NKT (GZMB$^+$NK1.1$^+$CD3$^+$) and (T) CD4$^+$ T cells (GZMB$^+$CD3$^+$CD4$^+$) in Tg and HbTg animals 10 days post Tg infection. N = 2–4 mice per group per experiment, 2 independent experiments. Data were tested for normality. ANOVA or Kruskal-Wallis tests were performed on parametric/non-parametric pooled data including N/Tg/HbTg/Hb groups, and when significant, Sidak's/Dunn's Multiple comparisons were performed on Naive vs. Tg and Tg vs. HbTg; n.s. = non significant.
(TIF)

**S7 Fig. Co-infected mice have similar levels of IFNγ and GZMB producing cells in the LIV.** 200 Hb larvae were given orally to mice 7 days prior to infection with 20 Tg tissue cysts. The cell percentage of IFNγ-producing (A) NK (IFNγ$^+$NK1.1$^+$CD3$^-$), (B) CD8$^+$ T (IFNγ$^+$CD3$^+$CD8$^+$), (C) γδ T (IFNγ$^+$CD3$^+$γδ+), (D) NKT (IFNγ$^+$NK1.1$^+$CD3$^+$) and (E) CD4$^+$ T cells (IFNγ$^+$CD3$^+$CD4$^+$) in Tg and HbTg animals 5 days post Tg infection. The percentage of IFNγ-producing (F) NK (IFNγ$^+$NK1.1$^+$CD3-), (G) CD8$^+$ T (IFNγ+CD3$^+$CD8$^+$), (H) γδ T (IFNγ$^+$CD3$^+$γδ+), (I) NKT (IFNγ$^+$NK1.1$^+$CD3$^+$) and (J) CD4$^+$ T cells (IFNγ$^+$CD3$^+$CD4$^+$) in Tg and HbTg animals 10 days post Tg infection. The cell percentage of GZMB-producing (K) NK (GZMB$^+$NK1.1$^+$CD3-), (L) CD8$^+$ T (GZMB$^+$CD3$^+$CD8$^+$), (M) γδ T (GZMB$^+$CD3$^+$γδ$^+$), (N) NKT (GZMB$^+$NK1.1$^+$CD3$^+$) and (O) CD4$^+$ T cells (GZMB$^+$CD3$^+$CD4$^+$) in Tg and HbTg animals 5 days post Tg infection. The cell percentage of GZMB-producing (P) NK (GZMB$^+$NK1.1$^+$CD3$^-$), (Q) CD8$^+$ T (GZMB$^+$CD3$^+$CD8$^+$), (R) NKT (GZMB$^+$NK1.1$^+$CD3$^+$), (S) γδ T (GZMB$^+$CD3$^+$γδ$^+$) and (T) CD4$^+$ T cells (GZMB$^+$CD3$^+$CD4$^+$) in Tg and HbTg animals 10 days post Tg infection. N = 2–4 mice per group per experiment, 2 independent experiments. Data were tested for normality. ANOVA or Kruskal-Wallis tests were performed on parametric/non-parametric pooled data including N/Tg/HbTg/Hb groups, and when significant, Sidak's/Dunn's Multiple comparisons were performed on Naive vs. Tg and Tg vs. HbTg; n.s. = non significant.
(TIF)

**S8 Fig. Single infected (Hb and Tg) and co-infected mice have similar numbers of MLN cells.** 200 Hb larvae were given orally to mice 7 days prior to infection with 20 Tg tissue cysts. Single cell suspensions were isolated from the mesenteric lymph nodes 5 days post Tg infection and viable cells were counted. Kruskal-Wallis tests were performed on non-parametric pooled data including N/Tg/HbTg/Hb groups, and when significant, Dunn's Multiple comparisons

were performed on Naive vs. Tg and Tg vs. HbTg; n.s. = non significant.
(TIF)

**S9 Fig. Co-infected mice have reduced numbers of MLN IFNγ and increased GZMB-producing cells 5 days post Tg infection.** 200 Hb larvae were given orally to mice 7 days prior to infection with 20 Tg tissue cysts. (A-E) The cell number of (A) NK (IFN©$^+$NK1.1$^+$CD3-), (B) CD8$^+$ T (IFN©$^+$CD3$^+$CD8$^+$), (C) ©δ T (IFN©$^+$CD3$^+$©δ$^+$), (D) NKT (IFN©$^+$NK1.1$^+$CD3$^+$) and (E) CD4$^+$ T cells (IFN©$^+$CD3$^+$CD4$^+$) in Tg and HbTg animals. (F-J) The cell number of IFN©-producing (F) NK (IFN©$^+$NK1.1$^+$CD3-), (G) CD8$^+$ T (IFN©$^+$CD3$^+$CD8$^+$), (H) ©δ T (IFN©$^+$CD3$^+$©δ$^+$), (I) NKT (IFN©$^+$NK1.1$^+$CD3$^+$) and (J) CD4$^+$ T cells (IFN©$^+$CD3$^+$CD4$^+$) in Tg and HbTg animals. (K-O) The cell number of GZMB-producing (K) NK (GZMB$^+$NK1.1$^+$CD3$^-$), (L) CD8$^+$ T (GZMB$^+$CD3$^+$CD8$^+$), (M) ©δ T (GZMB$^+$CD3$^+$©δ+), (N) NKT (GZMB$^+$NK1.1$^+$CD3$^+$) and (O) CD4$^+$ T cells (GZMB$^+$CD3$^+$CD4$^+$) in Tg and HbTg animals. N = 2–4 mice per group per experiment, 2 independent experiments. Data was tested for normality (Anderson-Darling test). ANOVA or Kruskal-Wallis tests were performed on parametric/non-parametric pooled data including N/Tg/HbTg/Hb groups, and when significant, Sidak's/Dunn's Multiple comparisons were performed on Hb vs. HbTg and Tg vs. HbTg; n.s. = non significant, * = $p<0.05$ and ** = $p<0.01$.
(TIF)

**S10 Fig. Pathology in the spleen and liver are similar between *T. gondii* and co-infected animals.** 200 Hb larvae were given orally to mice 7 days prior to infection with 20 Tg tissue cysts. Spleens and livers were harvested from mice 5 and 10 days post Tg infection, and formalin fixed. 6 μM slides were cut from the paraffin embedded swiss rolls, and stained with hematoxylin and eosin. Average spleen (A & B) and liver (C & D) pathology score for each mouse, euthanized at 5 (A &C) 10 (B & D) days post Tg infection. N>3 mice per group per experiment, 2 independent experiments. Data was tested for normality (Anderson-Darling test). ANOVA or Kruskal-Wallis tests were performed on parametric/non-parametric pooled data including N/Tg/HbTg/Hb groups, and when significant, Sidak's/Dunn's Multiple comparisons were performed on N vs Tg and Tg vs. HbTg; n.s. = non significant, * = $p<0.05$ and ** = $p<0.01$.
(TIF)

**S11 Fig. IFNγ serum protein levels do not differ between Tg and HbTg infected animals.** IFN© protein levels were measured in the serum by ELISA at 5 and 10 days post Tg infection. N = 2–4 mice per group per experiment, a minimum of 2 independent experiments. Data was tested for normality (Anderson-Darling test). ANOVA or Kruskal-Wallis tests were performed on parametric/non-parametric pooled data, and when significant, Sidak's/Dunn's Multiple comparisons were performed on Hb vs. HbTg and Tg vs. HbTg; **** = $p<0.0001$.
(TIF)

**S1 Table. Small intestine pathology scales.** The total score for each animal was calculated by adding the score for the pathology associated with epithelial integrity and inflammatory cell infiltrate.
(XLSX)

**S2 Table. Spleen and liver pathology scales.** For the spleen, the total score for each animal was calculated by adding the score for the pathology associated with the follicle shape, the marginal zone and the germinal centre. For the liver, the total score for each animal was calculated by adding the score for the pathology associated with necrosis, the infiltrate size, the

proportion of infiltrates and the nature of the perivascular infiltrates.
(XLSX)

## Acknowledgments

We would like to thank the University of Calgary LESARC Facility technicians, especially Dawn Martin for their continued support as well as the University of Calgary's Diagnostic Services Unit, especially Sue Calder-Lodge and JJ Larios for their patience and expertise.

## Author Contributions

**Conceptualization:** Edina K. Szabo, Constance A. M. Finney.

**Data curation:** Edina K. Szabo, Constance A. M. Finney.

**Formal analysis:** Edina K. Szabo, Emma Forrester, Holly Liu, Beverly Dong, Aralia Leon Coria, Shashini Perera, Namratha Badawadagi, Camila Gaio, Constance A. M. Finney.

**Funding acquisition:** Constance A. M. Finney.

**Investigation:** Edina K. Szabo, Christina Bowhay, Emma Forrester, Holly Liu, Beverly Dong, Aralia Leon Coria, Shashini Perera, Beatrice Fung, Namratha Badawadagi, Camila Gaio, Kayla Bailey, Manfred Ritz, Constance A. M. Finney.

**Methodology:** Edina K. Szabo, Christina Bowhay, Emma Forrester, Holly Liu, Beverly Dong, Aralia Leon Coria, Shashini Perera, Beatrice Fung, Namratha Badawadagi, Camila Gaio, Kayla Bailey, Manfred Ritz, Joel Bowron, Anupama Ariyaratne, Constance A. M. Finney.

**Project administration:** Edina K. Szabo, Constance A. M. Finney.

**Resources:** Constance A. M. Finney.

**Supervision:** Edina K. Szabo, Constance A. M. Finney.

**Validation:** Edina K. Szabo, Christina Bowhay, Constance A. M. Finney.

**Visualization:** Edina K. Szabo, Constance A. M. Finney.

**Writing – original draft:** Edina K. Szabo, Constance A. M. Finney.

**Writing – review & editing:** Edina K. Szabo, Constance A. M. Finney.

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
