## [Decision Letter · Decision Letter 0]

19 Jan 2024

PONE-D-23-30525Heligmosomoides bakeri and Toxoplasma gondii co-infection leads to increased mortality associated with intestinal pathologyPLOS ONE

Dear Dr. Finney,

Thank you for submitting your manuscript to PLOS ONE. After careful consideration, we feel that it has merit but does not fully meet PLOS ONE’s publication criteria as it currently stands. Therefore, we invite you to submit a revised version of the manuscript that addresses the points raised during the review process.

We look forward to receiving your revised manuscript.

Kind regards,

Subash Babu

Academic Editor

PLOS ONE

Journal Requirements:

"..This works was funded through Dr. Finney’s grants from the Canadian Foundation for Innovation and the Natural Sciences and Engineering Research Council of Canada (NSERC), as well as scholarships for Dr Anupama Ariyaratne (NSERC Create in Host Parasite Interactions), Namratha Badawadagi (University of Calgary Markin scholarship), Kayla Bailey and Emma Forrester (University of Calgary PURE Scholarship), Dr Joel Bowron and Beverly Dong (NSERC), Camila Gaio and Manfred Ritz (Mitacs Globalinks Scholarships), Shashini Perera (Alberta Graduate Excellence Scholarship) and Dr Edina K Szabo (UCalgary Eyes High Postdoctoral Scholarship)."

Funding information should not appear in the Acknowledgments section or other areas of your manuscript. We will only publish funding information present in the Funding Statement section of the online submission form. 

"This works was funded through Dr. Finney’s grants from the Canadian Foundation for Innovation and the Natural Sciences and Engineering Research Council of Canada (NSERC), as well as scholarships for Dr Anupama Ariyaratne (NSERC Create in Host Parasite Interactions), Namratha Badawadagi (University of Calgary Markin scholarship), Kayla Bailey and Emma Forrester (University of Calgary PURE Scholarship), Dr Joel Bowron and Beverly Dong (NSERC), Camila Gaio and Manfred Ritz (Mitacs Globalinks Scholarships), Shashini Perera (Alberta Graduate Excellence Scholarship) and Dr Edina K Szabo (UCalgary Eyes High Postdoctoral Scholarship). 

5. Please upload a new copy of Supporting Figure S5 as the detail is not clear. Please follow the link for more information: 

https://blogs.plos.org/plos/2019/06/looking-good-tips-for-creating-your-plos-figures-graphics/

https://blogs.plos.org/plos/2019/06/looking-good-tips-for-creating-your-plos-figures-graphics/

Reviewers' comments:

Reviewer's Responses to Questions

**Comments to the Author**

1. Is the manuscript technically sound, and do the data support the conclusions?

Reviewer #1: Partly

Reviewer #2: Yes

2. Has the statistical analysis been performed appropriately and rigorously? 

Reviewer #1: Yes

Reviewer #2: Yes

3. Have the authors made all data underlying the findings in their manuscript fully available?

Reviewer #1: Yes

Reviewer #2: No

4. Is the manuscript presented in an intelligible fashion and written in standard English?

Reviewer #1: Yes

Reviewer #2: Yes

5. Review Comments to the Author

Reviewer #1: The manuscript title “Heligmosomoides bakeri and Toxoplasma gondii co-infection leads to increased mortality associated with intestinal pathology” by Szabo EK et al., aimed to investigate the impact of different cytokine producing lymphocyte population on parasite control/clearance. These analyses are reasonably well done and novelty of the study is sufficient.

Major comments:

Line 261-262: Here, the authors have mentioned that increased mortality seen in co-infected group was due to the intestinal worm burden only. But, in fig1, except MLN Tg burden in HbTg group at day 10 showed significant differences, but none of the other parameters like granuloma numbers and size and macrophage counts were not showing any differences between the compared groups. It is not clear why the authors think the mortality of animals were due to the Tg burden only, but not other factors? For instance, the mortality could be due to contribution of Tg mediated overt Th1 immune responses and also infiltration of other inflammatory cells at day 10 post infection? Please clarify.

Line 297-298: In Fig 1C, it looks like Hb burden from two mice in co-infection group are outliers and that could result significance differences between these two comparisons with p<0.05. Did authors looked the same results even after removing these outliers? if not, please rephrase the sentence here. Also, the error bars at day10 in Fig 1F seems to be no differences between the groups. Please double check the significance here.

Line 310-314: It seems that figure numbers are mislabeled here. Double check all the figure numbers.

Line 367: Please separate the interpretations for Il-4 and Il-13 expression in the text, because Il-4 expression was not increased even in day 5 in Hb infected animals, but only at day10, both Il-4 and Il-13 expression were increased. Also, rephrase the sentence here by mentioning the differences in the day 5 and day 10 for each cytokine.

Line 369-370: In the example here, only Il-13 expression was increased, but not Il-4 at day 5pi.

Line 387 (Fig 3A & 3F): In Fig 3A, it looks like CD4 producing Ifng levels were almost null in all 4 groups, but in Fig 3F, it was shown as CD4 expressing Ifng from Tg group is significantly different than co-infected group. Are these subtle differences coming from the isolated MLN or from whole intestine?

Line 471-473: It seems to be that figure numbers are mislabeled here. There is no 5J, 5K-O in the attached main figures.

Line 491-493: Here authors have mentioned that “differences in mortality (Fig 1B)” we saw no differences….. But in Fig 1B, the actual differences in mortality between Tg and HbTg starts at day 8, but not at day 5. Hence, we don’t expect any differences between the groups at day 5, but only at day 10. Therefore, day 10 data should be ideal to include as representative figure in the place of day 5, although there were no differences noted at day 10 (as mentioned by authors in the text “data not shown”).

Line 497-506 (In Fig 6): There is no differences seen between Tg and HbTg in the majority of the parameters tested here, except Fig 6H, 6I and 6L. Therefore, the title for this figure description is slightly overstated. If not the inflammation and also Tg burden (partially), then who is contributing the mortality of the animal? Did authors look into the other non-lymphoid compartments in parallel, especially the role of inflammatory monocytes and neutrophils for the death of the animal with necrotic tissues? Although we understand that the aims of this manuscript is to investigate the cytokine expression in the lymphoid compartment, but the parallel lack of interrogation on the contribution of non-lymphoid immune cells is evident. This has become important as the manuscript titled “co-infection leads to increased mortality with intestinal pathology”. Therefore, it is suggested to mention and discuss the potential effects of overt pathology by other immune cells in addition to the lymphoid compartment that could possibly led to mortality (in the limitation of the study section).

Reviewer #2: The research aims to elucidate the effect of a co-infection, which the parasites induce opposite immune responses, on the parasite control and clearance. Overall the paper is well written, the figures are understandable, and the conclusions do not extrapolate the results. There are two major concerns: 1. Timeline for the co-infection seems not to be the appropriate one for the aim of the research, and 2. The authors should pay more attention in the figure legends, reference of the figures along the paper and the figures that they submitted. There are several mistakes.

Please address the comments described below:

1. Can the authors explain why they chose 7 days post Hb infection to do the Tg co-infection? 7 days is the worst timing for Hb infection. It's not a chronic infection yet, the immune response in the intestine is not established. Why not wait a couple of weeks with Hb infection to then infect with Tg? Or why use a C57Bl/6 mice that is not going to give a strong Th2 response and also will have the worst pathology?

2. The content written in lines 63 to 70 does not help the point of the research. Inserting this information at that point of introduction is confusing and does not connect with the following paragraphs. Review the necessity of that information, or consider including it after line 105.

3. Consider adding a conclusion to the introduction section.

4. Can the authors comment on the necessity of stimulating the cells for Flow Cytometry with PMA and Ionomycin? Since the animals are already infected with two pathogens, why stimulated the cells for many hours with a polyclonal stimuli? Wouldn't a non-stimulated approach or a stimulation with parasite antigen be a better approach to the research goal?

5. Line 271 - Instead of 16 days post Tg infection, should be Hb infection.

6. Between lines 304 and 314 all references of the figures are wrong. For example, line 304 reference figures 1E & 1F, but it should be 1F & 1G.

7. Figure 1B - Shouldn't it be Days post Hb infection? Also, the reader can't see the survival line for the naive mice.

8. In Figure 1 when the authors add the timepoints of D5 and D10 in the x axis, specially bellow Hb data, they are confusing the reader. Considering Hb infection the timepoints should be D12 and D17.

9. Figure 1F - There is a typo in the group names. Hp instead of Hb.

10. The authors should evaluate the need of using dots and bars. In some figures like 1F, the presence of the bars can mislead the reader to think that there is no difference between the groups or that there is an increase in the HpTg group.

11. Figure 2: Considering the Th2 cytokines, showing the comparison between Hb and HbTg be more useful for the research, since the Hb infection is the one inducing this cytokines?

12. Could the lower levels of IFN in the HbTg group (D5) be related to the actual increase in the levels of Th2 cytokines for the same timepoint? Can the author comment about that?

13. Do the authors think it's relevant for the paper to show the levels of just IFN in the blood? It makes the reader curious about the systemic level of all cytokines, and we don't see this result.

14. Back to PMA stimulation, do the authors think that the levels of IFN/GranzymeB measured by flow could be misrepresented since the cells could be exhausted?

15. Where are Figures 5J to 5O?

16. Line 468 - Since there is a difference in the MLN number of cells it would be important to support the data from Figures 3 and 4, to show the difference and also the analysis considering the number of cells and not only the %.

17. Considering the results presented in Figure 6 do the authors think that the tittle of the paper is appropriate? Is the increase in mortality associated with pathology, since there is no difference between HbTg and Hb?

18. General comment about the results: It could be useful to expand/show the analysis considering all the groups and timepoints. For example, we don't see the comparison with the groups and the naive mice in all figures, also the analysis between the co-infection group considering the differences between D5 and D10 could increase the depth of the results.

6. PLOS authors have the option to publish the peer review history of their article (what does this mean?). If published, this will include your full peer review and any attached files.

Reviewer #1: No

Reviewer #2: No

---

## [Author Response · Author response to Decision Letter 0]

3 Mar 2024

We wish to thank the reviewers and editor for taking the time to read and review our manuscript. Time for these efforts is difficult to find, and we value your input. We appreciate all the insightful comments which have helped create a better version of our original submission. Please see our detailed response below. 

Journal Requirements:

We have changed all file names and updated the bibliography and affiliations to fit with Plos One’s requirements.

We have provided grant numbers when they exist. For those that do not have external grant numbers (from the funding agency), we have not provided a number. 

"..This works was funded through Dr. Finney’s grants from the Canadian Foundation for Innovation and the Natural Sciences and Engineering Research Council of Canada (NSERC), as well as scholarships for Dr Anupama Ariyaratne (NSERC Create in Host Parasite Interactions), Namratha Badawadagi (University of Calgary Markin scholarship), Kayla Bailey and Emma Forrester (University of Calgary PURE Scholarship), Dr Joel Bowron and Beverly Dong (NSERC), Camila Gaio and Manfred Ritz (Mitacs Globalinks Scholarships), Shashini Perera (Alberta Graduate Excellence Scholarship) and Dr Edina K Szabo (UCalgary Eyes High Postdoctoral Scholarship)."

Funding information should not appear in the Acknowledgments section or other areas of your manuscript. We will only publish funding information present in the Funding Statement section of the online submission form. 

Please remove any funding-related text from the manuscript and let us know how you would like to update your Funding Statement. 

We have removed this section. 

Currently, your Funding Statement reads as follows: 

"This works was funded through Dr. Finney’s grants from the Canadian Foundation for Innovation and the Natural Sciences and Engineering Research Council of Canada (NSERC), as well as scholarships for Dr Anupama Ariyaratne (NSERC Create in Host Parasite Interactions), Namratha Badawadagi (University of Calgary Markin scholarship), Kayla Bailey and Emma Forrester (University of Calgary PURE Scholarship), Dr Joel Bowron and Beverly Dong (NSERC), Camila Gaio and Manfred Ritz (Mitacs Globalinks Scholarships), Shashini Perera (Alberta Graduate Excellence Scholarship) and Dr Edina K Szabo (UCalgary Eyes High Postdoctoral Scholarship). 

This has been included in the cover letter.

All instances of ‘data not shown’ have been replaced with Supporting Figures.

5. Please upload a new copy of Supporting Figure S5 as the detail is not clear. 

Figure has been amended and uploaded. 

Reviewers' comments:

Major comments:

Line 261-262: Here, the authors have mentioned that increased mortality seen in co-infected group was due to the intestinal worm burden only. But, in fig1, except MLN Tg burden in HbTg group at day 10 showed significant differences, but none of the other parameters like granuloma numbers and size and macrophage counts were not showing any differences between the compared groups. It is not clear why the authors think the mortality of animals were due to the Tg burden only, but not other factors? For instance, the mortality could be due to contribution of Tg mediated overt Th1 immune responses and also infiltration of other inflammatory cells at day 10 post infection? Please clarify.

Apologies for the confusion. Increased mortality at this point in the paper (Fig 1) is being studied in the context of parasite burden (Tg and Hp). We have removed the graphs relating to granuloma cell composition and added them later to the intestinal pathology figures (6 & 7) for clarity. In this later section, we now highlight cell infiltration on day 10 post Tg infection as being associated with HbTg infected animals (lines 545-546). 

Line 297-298: In Fig 1C, it looks like Hb burden from two mice in co-infection group are outliers and that could result significance differences between these two comparisons with p<0.05. Did authors looked the same results even after removing these outliers? if not, please rephrase the sentence here.

For Fig 1C, data in all 4 groups are normally distributed according to the D’Agostino Pearson normality test. 

Hb burden is variable. Please see the following as way of example: DOI:10.3791/52412, DOI: 10.1084/jem.20101074, DOI: 10.3389/fimmu.2022.1020056

If we remove both data points and rerun the analysis, the data remain significant. 

Also, the error bars at day10 in Fig 1F seems to be no differences between the groups. Please double check the significance here.

We have double checked significance. Our values are correct. To better visualise the difference, we have amended the colours on the graph. 

Line 310-314: It seems that figure numbers are mislabeled here. Double check all the figure numbers.

We have amended the incorrect figure numbers. 

Line 367: Please separate the interpretations for Il-4 and Il-13 expression in the text, because Il-4 expression was not increased even in day 5 in Hb infected animals, but only at day10, both Il-4 and Il-13 expression were increased. Also, rephrase the sentence here by mentioning the differences in the day 5 and day 10 for each cytokine.

We have now separated the interpretations and mentioned all differences between IL-4 and IL-13 at the 2 different time points (lines 359-372). 

Line 369-370: In the example here, only Il-13 expression was increased, but not Il-4 at day 5pi.

We have now separated the interpretations and mentioned all differences between IL-4 and IL-13 at the 2 different time points (lines 359-372).

Line 387 (Fig 3A & 3F): In Fig 3A, it looks like CD4 producing Ifng levels were almost null in all 4 groups, but in Fig 3F, it was shown as CD4 expressing Ifng from Tg group is significantly different than co-infected group. Are these subtle differences coming from the isolated MLN or from whole intestine?

The data in figure 3 are all from the MLN. This is mentioned in the figure legend and title. We have no flow cytometry data from the intestine since obtaining live lymphocytes for PMA/ionomycin stimulation from parasite-infected intestines proved unreliable. However, the data from the PP (closely linked to the SI) show no difference in IFNg producing CD4 cells, as stated in the text.

The Y axis in Fig 3F has also been changed to better visualise the difference between the HbTg and Tg groups.

Line 471-473: It seems to be that figure numbers are mislabeled here. There is no 5J, 5K-O in the attached main figures.

Apologies. This was supposed to be labelled SI Fig 5. It has been relabelled appropriately. 

Line 491-493: Here authors have mentioned that “differences in mortality (Fig 1B)” we saw no differences….. But in Fig 1B, the actual differences in mortality between Tg and HbTg starts at day 8, but not at day 5. Hence, we don’t expect any differences between the groups at day 5, but only at day 10. Therefore, day 10 data should be ideal to include as representative figure in the place of day 5, although there were no differences noted at day 10 (as mentioned by authors in the text “data not shown”).

Although differences in mortality only start at day 8, we would expect immunological/pathology changes before that point that would lead to these differences. For example, in https://doi.org/10.4049/jimmunol.2200504 , a very similar study, they see mortality from day 12 post infection. However, they study immunity prior to mortality (day 3-10). Specifically, they look at IFNg producing cells at day 10, ~2 days before the onset of mortality. This equates to our day 5 time point. Other publications show that Tg infection alone induces an increase in IL-12 and IFNg from day 3 onwards (PMID: 7915739 and doi: 10.1371/journal.ppat.1009970). We were therefore interested in the day 5 time point. 

We have added day 10 data (Fig 7) and changed the wording in this section to make the importance of both time points clearer (see section: HbTg infected animals have intestinal pathology characteristic of both Tg and Hb parasites). 

Line 497-506 (In Fig 6): There is no differences seen between Tg and HbTg in the majority of the parameters tested here, except Fig 6H, 6I and 6L. Therefore, the title for this figure description is slightly overstated. If not the inflammation and also Tg burden (partially), then who is contributing the mortality of the animal? Did authors look into the other non-lymphoid compartments in parallel, especially the role of inflammatory monocytes and neutrophils for the death of the animal with necrotic tissues? Although we understand that the aims of this manuscript is to investigate the cytokine expression in the lymphoid compartment, but the parallel lack of interrogation on the contribution of non-lymphoid immune cells is evident. This has become important as the manuscript titled “co-infection leads to increased mortality with intestinal pathology”. Therefore, it is suggested to mention and discuss the potential effects of overt pathology by other immune cells in addition to the lymphoid compartment that could possibly led to mortality (in the limitation of the study section).

Unfortunately, we did not study non lymphoid cells. However, others have (https://doi.org/10.4049/jimmunol.2200504 ) and shown that in a similar infection model to ours (200 Hb 9 days prior to 10 cysts Me49 Tg), early infiltration of inflammatory monocytes and neutrophils to the peritoneal cavity was impaired in HbTg mice and that this may ultimately account for the reduced IFNg production by lymphoid cells. This dysregulation was later restored and did not solely account for differences in mortality. 

We have now added text to reflect this and changed the title of the section to clarify our findings (lines 608-610).

Reviewer #2: 

1. Can the authors explain why they chose 7 days post Hb infection to do the Tg co-infection? 7 days is the worst timing for Hb infection. It's not a chronic infection yet, the immune response in the intestine is not established. Why not wait a couple of weeks with Hb infection to then infect with Tg? Or why use a C57Bl/6 mice that is not going to give a strong Th2 response and also will have the worst pathology?

From previous publications we know that co-infection outcome can vary depending on infection time, length and the order of which parasitic infection established itself first (DOI: 10.3389/fcimb.2017.00341). We specifically wanted to focus on the acute infection phases of both parasites for the following reasons:

- Many studies on HbTg co-infection infect animals with Tg as Hb adults emerge and start to reproduce, between 7 and 12 days post Hb infection: doi.org/10.4049/jimmunol.2200504 , DOI: 10.1128/IAI.01236-07, doi: 10.1128/IAI.73.9.5468-5481.2005 , doi: 10.1016/j.cell.2021.01.051, https://doi.org/10.4049/jimmunol.1601741. To compare our results to theirs, we stayed within this timeframe. 

- At day 7 post-infection, Hb specific Th2 responses (IL-4 and IL-13) are already measurable in the MLN, where Treg numbers are also significantly increased (DOI: 10.1002/eji.201948392). Granulomas are also visible in the intestinal mucosa and genes involved in monocyte and eosinophil attraction as well as ECM remodelling are increased from 20 to >4000 fold (DOI 10.3389/fimmu.2022.1020056). 

- Our model represents the situation when people/animals are infected for the first time. While chronic infection offers interesting insights, it is unlikely that people/animals living in endemic regions are infected chronically just once. They are likely infected multiple times while they remain in the same environment (grazing/trickle model – well described for Trichuris muris here: https://doi.org/10.1371/journal.ppat.1007926 and for H. bakeri here DOI 10.3389/fimmu.2022.1020056 and/or drug cure/reinfection model). The impact of these more realistic scenarios at later time points is something of great interest to the group but beyond the scope of this study. 

We are using C57BL/6 for the following reasons:

- From https://doi.org/10.1016/j.pt.2016.11.007: ‘BALB/c mice are less susceptible to infection by oral inoculation with cysts. Brain cysts, a marker of dissemination and chronicity, are either not formed or present in small numbers in BALB/c mice [47,49,57,58].’

- Chronic Hb infection establishes itself so it represents a good model for susceptible individuals.

2. The content written in lines 63 to 70 does not help the point of the research. Inserting this information at that point of introduction is confusing and does not connect with the following paragraphs. Review the necessity of that information, or consider including it after line 105.

The information has been moved. 

3. Consider adding a conclusion to the introduction section.

A conclusion has been added. 

4. Can the authors comment on the necessity of stimulating the cells for Flow Cytometry with PMA and Ionomycin? Since the animals are already infected with two pathogens, why stimulated the cells for many hours with a polyclonal stimuli? Wouldn't a non-stimulated approach or a stimulation with parasite antigen be a better approach to the research goal?

At the start of the project, we optimised all techniques. We first started with no polyclonal stimulus but could not obtain strong reproducible responses. We reverted to using PMA/Ionomycin.

5. Line 271 - Instead of 16 days post Tg infection, should be Hb infection.

The text is correct. For survival, days are measured post Tg infection.

6. Between lines 304 and 314 all references of the figures are wrong. For example, line 304 reference figures 1E & 1F, but it should be 1F & 1G.

We have amended the incorrect figure numbers. 

7. Figure 1B - Shouldn't it be Days post Hb infection? Also, the reader can't see the survival line for the naive mice.

The text is correct. For survival, days are measured post Tg infection.

The colours have been amended to see both naïve and Hb 100% survival. 

8. In Figure 1 when the authors add the timepoints of D5 and D10 in the x axis, specially bellow Hb data, they are confusing the reader. Considering Hb infection the timepoints should be D12 and D17. 

We have amended the numbers for the graphs relating to Hb parameters. 

9. Figure 1F - There is a typo in the group names. Hp instead of Hb.

We have corrected the typo.

10. The authors should evaluate the need of using dots and bars. In some figures like 1F, the presence of the bars can mislead the reader to think that there is no difference between the groups or that there is an increase in the HpTg group.

The colours and bars have been amended to make it easier to interpret the figures. 

11. Figure 2: Considering the Th2 cytokines, showing the comparison between Hb and HbTg be more useful for the research, since the Hb infection is the one induc

---

## [Decision Letter · Decision Letter 1]

14 Mar 2024

Heligmosomoides bakeri and Toxoplasma gondii co-infection leads to increased mortality associated with changes in immune resistance in the lymphoid compartment and disease pathology

PONE-D-23-30525R1

Dear Dr. Finney,

We’re pleased to inform you that your manuscript has been judged scientifically suitable for publication and will be formally accepted for publication once it meets all outstanding technical requirements.

Kind regards,

Subash Babu

Academic Editor

PLOS ONE

Additional Editor Comments (optional):

Reviewers' comments:

Reviewer's Responses to Questions

**Comments to the Author**

1. If the authors have adequately addressed your comments raised in a previous round of review and you feel that this manuscript is now acceptable for publication, you may indicate that here to bypass the “Comments to the Author” section, enter your conflict of interest statement in the “Confidential to Editor” section, and submit your "Accept" recommendation.

Reviewer #1: All comments have been addressed

Reviewer #2: All comments have been addressed

2. Is the manuscript technically sound, and do the data support the conclusions?

Reviewer #1: (No Response)

Reviewer #2: Yes

3. Has the statistical analysis been performed appropriately and rigorously? 

Reviewer #1: Yes

Reviewer #2: Yes

4. Have the authors made all data underlying the findings in their manuscript fully available?

Reviewer #1: Yes

Reviewer #2: Yes

5. Is the manuscript presented in an intelligible fashion and written in standard English?

Reviewer #1: Yes

Reviewer #2: Yes

6. Review Comments to the Author

Reviewer #1: The authors have revised the manuscript substantially, and answered all my queries with justification and made required changes in the text as well as in the figures. Therefore, I recommend to accept this manuscript after addressing the following 2 minor comments. Thank You

1. Line 112-113 (Revised manuscript): These lines should either be removed (or) carefully stated, as it may project adversely for the results coming out from the researchers who are working on single infection models.

2. Line 375 (Revised manuscript): Result title for figure 3 & 4 is misleading and needs to be revised. The authors have mentioned that there is a “decreased granzyme B production” in the title. But in Fig 4, granzyme B production was not significantly decreased (Fig 4A-F), instead there is an increased production of granzyme B (Fig 4G-L) (Line 412 – 413).

Reviewer #2: (No Response)

7. PLOS authors have the option to publish the peer review history of their article (what does this mean?). If published, this will include your full peer review and any attached files.

Reviewer #1: No

Reviewer #2: No

---

## [Editor Report · Acceptance letter]

29 Apr 2024

PONE-D-23-30525R1 

PLOS ONE

Dear Dr. Finney, 

I'm pleased to inform you that your manuscript has been deemed suitable for publication in PLOS ONE. Congratulations! Your manuscript is now being handed over to our production team.

Kind regards, 

on behalf of

Dr. Subash Babu 

Academic Editor

PLOS ONE